# Inducible auto-phosphorylation regulates a widespread family of nucleotidyltransferase toxins

Tom J. Arrowsmith[1], Xibing Xu[2], Shangze Xu[3], Ben Usher[1], Peter Stokes[4], Megan Guest[1], Agnieszka K. Bronowska[3], Pierre Genevaux[2]✉ & Tim R. Blower[1]✉

Nucleotidyltransferases (NTases) control diverse physiological processes, including RNA modification, DNA replication and repair, and antibiotic resistance. The *Mycobacterium tuberculosis* NTase toxin family, MenT, modifies tRNAs to block translation. MenT toxin activity can be stringently regulated by diverse MenA antitoxins. There has been no unifying mechanism linking antitoxicity across MenT homologues. Here we demonstrate through structural, biochemical, biophysical and computational studies that despite lacking kinase motifs, antitoxin $MenA_1$ induces auto-phosphorylation of $MenT_1$ by repositioning the $MenT_1$ phosphoacceptor T39 active site residue towards bound nucleotide. Finally, we expand this predictive model to explain how unrelated antitoxin $MenA_3$ is similarly able to induce auto-phosphorylation of cognate toxin $MenT_3$. Our study reveals a conserved mechanism for the control of tuberculosis toxins, and demonstrates how active site auto-phosphorylation can regulate the activity of widespread NTases.

Nucleotidyltransferases (NTases) are ubiquitous and abundant throughout nature[1,2]. NTases have roles in RNA processing and modification[3], genomic replication, repair, and remodelling[2], and antibiotic resistance[4]. NTase superfamilies share high structural similarity owing to conserved catalytic motifs and overall NTase fold topology[1] but lack homology at the sequence level as a result of divergence in their biological targets[2]. By necessity, organisms have evolved efficient modes of regulation to ensure NTase activity can be stringently controlled. These include negative feedback inhibition by reaction products or intermediates[5], reversal of RNA processing by 3′ ribonucleases[6], and covalent modifications to the NTase domain that disrupt interactions with target substrates[7]. DUF1814-family NTases were identified as widespread throughout bacteria, archaea, and fungi, and include the MenAT toxin-antitoxin (TA) systems[6,8].

Like NTases, TA systems are ubiquitous within bacterial and archaeal genomes alike and are oftentimes encoded on mobile genetic elements[9,10]. TA systems are involved in diverse processes including bacteriophage defence, virulence, plasmid addiction, and stress responses[11–14]. A new study has shown that some TA systems may also have a role in bacterial physiology, through the discovery of hallmark toxin modifications being generated during regular bacterial growth[15]. Paradigmatic TA systems are small bipartite modules encoding a growth-inhibitory toxic protein, responsible for targeting an essential intracellular pathway, and an antagonistic antitoxin capable of neutralizing the toxin[16]. By necessity, bacteria have evolved diverse means to provide exquisite control over toxin activity levels to fit current physiological need[16]. *Mycobacterium tuberculosis*, the causative agent of human tuberculosis, harbours an unusually large arsenal of TA systems[17], many of which have been linked to pathogenicity, potentially aiding immune evasion and antibiotic resistance[18,19]. The MenAT family of conserved NTase toxins and their cognate antitoxins represent four TA systems from *M. tuberculosis* (Fig. 1a)[6,20,21]. $MenT_1$ and

[1]Department of Biosciences, Durham University, Durham, UK. [2]Laboratoire de Microbiologie et Génétique Moléculaires (LMGM), Centre de Biologie Intégrative (CBI), Université de Toulouse, CNRS, Université Toulouse III - Paul Sabatier (UT3), Toulouse, France. [3]Chemistry – School of Natural and Environmental Sciences, Newcastle University, Newcastle Upon Tyne, UK. [4]Department of Chemistry, Durham University, Durham, UK. ✉e-mail: pierre.genevaux@univ-tlse3.fr; timothy.blower@durham.ac.uk

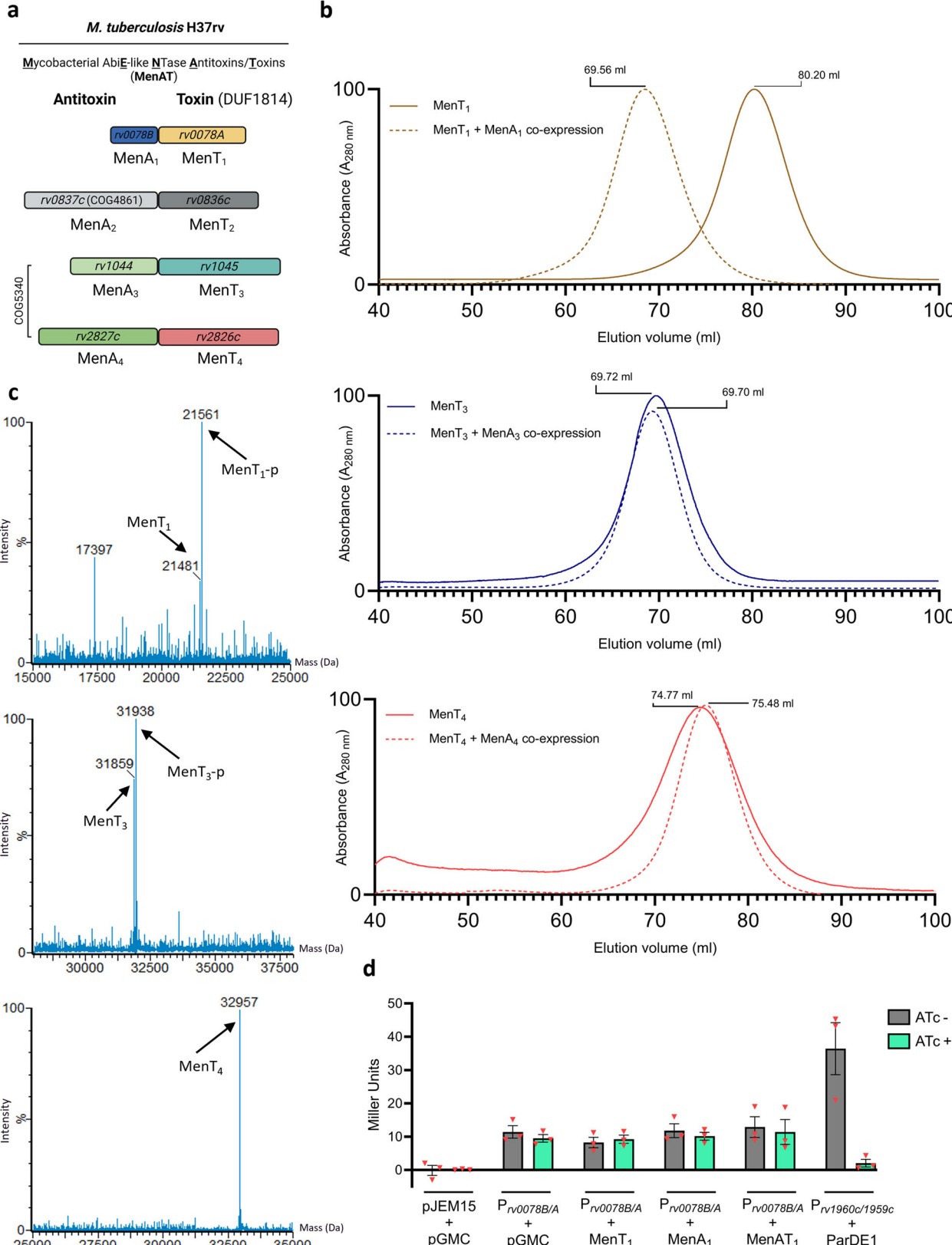

**Fig. 1 | MenAT systems differ in their modes of antitoxicity and regulation.**
**a** Scaled representation of the *M. tuberculosis* MenAT TA systems with original gene identifiers and revised nomenclature for gene products. **b** Top to bottom; overlaid SEC traces of MenT$_1$, MenT$_3$, and MenT$_4$ expressed and purified in the absence and presence of cognate MenA antitoxins. Chromatograms are normalized between 0 and 100 for presentation and comparison, cropped to the appropriate scale. Samples were analysed using a HiPrep™ 16/60 Sephacryl® S-200 HR column. **c** ES⁺-ToF MS of purified MenT$_1$, MenT$_3$, and MenT$_4$ expressed in the presence of cognate MenA antitoxins. **d** β-galactosidase activity of *M. smegmatis* mc²–155 co-transformed with pJEM15 -vector, pJEM15-P$_{rv0078B/A}$, or pJEM15-P$_{rv1960c/1959c}$, and either pGMC -vector, pGMC -MenT$_1$, -MenA$_1$, -MenAT$_1$, or −ParDE1. Data are representative of three independent biological replicates and bars display mean values +/- SEM.

$MenT_3$ have been shown to transfer pyrimidines to 3' CCA tRNA acceptor stems and block translation, with $MenT_1$ displaying greater promiscuity for target tRNAs and nucleotide substrates than $MenT_3$[6,20]. Of the four MenT toxins, $MenT_1$ and $MenT_3$ have so far been shown to inhibit growth of *M. tuberculosis* when expressed in the absence of their cognate antitoxins[6,20]. Both $MenA_3$ and $MenA_4$ were previously identified as essential for growth of the pathogen in vivo[22], suggesting their cognate toxins inhibit *M. tuberculosis* growth, and an *M. tuberculosis* Δ*menT2* mutant also recently showed an infection defect in guinea pigs[6,23].

MenT toxins are homologues of AbiEii from AbiE, the fourth most abundant bacteriophage defence system in prokaryotic organisms[24]. MenT and AbiEii toxins share four highly conserved NTase motifs scattered throughout their structures (Supplementary Fig. 1A), proposed to facilitate co-ordination of metals (motifs I and II), base stacking, and transfer of nucleotides to tRNA isoacceptors (motif III), and to contribute to catalysis (motif IV)[2,8]. Unlike the MenT toxins, the four MenA antitoxins are diverse, representing three independent protein families[6] (Fig. 1a). This suggests that functionally divergent modes of antitoxicity might exist between each system. Recent characterization of the $MenAT_1$ and $MenAT_3$ TA systems found that $MenA_1$ binds asymmetrically to two $MenT_1$ toxin protomers to form a stable heterotrimeric complex in vitro[20], whereas $MenA_3$ appears to function as a kinase, phosphorylating $MenT_3$ at the highly conserved catalytic S78[6,21]. In contrast, comparably little is known with regards to the mode of antitoxicity employed by $MenA_4$ to inhibit $MenT_4$, though $MenA_4$ is a homologue of $MenA_3$ and AbiEi antitoxins[25,26].

In this work, we provide structural, functional, biophysical, and computational characterization of MenAT systems in order to better understand NTases regulation. Biochemical and X-ray crystallographic analyses show that, like $MenT_3$, $MenT_1$ is also phosphorylated in the presence of its cognate antitoxin, though the antitoxins are structurally unrelated. This is particularly unusual as $MenA_1$ interacts with $MenT_1$ via a single extended α-helix. Further biophysical analyses led to atomistic molecular dynamics simulations of complex formation in the presence of nucleotide substrate. Our calculated results suggest that MenT toxins can auto-phosphorylate to control activity, catalysed by antitoxin-dependent movement of the target active site loop residue towards the donor phosphate. This mechanism represents a conserved molecular method for the control of tuberculosis toxins, as representatives of the widespread DUF1814 family. The data provide further insights into the control of translation within a deadly bacterial pathogen and indicates potential routes for regulation across NTases.

## Results

### MenAT systems differ in their modes of antitoxicity and regulation

We previously hypothesized that the lack of structural homology between MenA antitoxins indicated differing modes of antitoxicity were used to regulate MenT homologues[20]. To explore this hypothesis, we expressed and purified $MenT_1$, $MenT_3$ and $MenT_4$ in the absence and presence of cognate MenA antitoxins ($MenT_2$ was omitted due to a lack of detectable activity[6]), and analysed each sample using size exclusion chromatography (SEC) (Fig. 1b). Despite demonstrable toxicity of MenT NTases, high yields of each lone toxin homologue (2–5 mg/L) can be obtained following recombinant protein expression in *E. coli* using rich media, which we previously suggested may be a result of elevated tRNA target levels in *E. coli* relative to *M. tuberculosis*[20]. Co-expression of $MenAT_1$ resulted in production of the heterotrimeric $MenT_1α$:$MenA_1$:$MenT_1β$ complex in comparison to $MenT_1$ alone (Fig. 1b; Supplementary Fig. 1), which we have previously characterized[20]. The elution profiles following co-expression of $MenAT_3$ or $MenAT_4$ did not change compared to those of corresponding toxins expressed alone (Fig. 1b). Previous structural characterization of $MenT_3$ and $MenT_4$ following co-expression with their

cognate antitoxins ($MenT_3$, PDB 6Y5U[6] PDB 6J7S[21]; $MenT_4$, PDB 6Y56[6]) revealed a phosphoserine in $MenT_3$, but not $MenT_4$. To confirm whether post-translational modifications had been made to any of the toxins in the presence of each antitoxin, we analysed each purified expression sample by electrospray time-of-flight ($ES^+$-ToF) mass spectrometry (Fig. 1c; Supplementary Fig. 1B–D). In agreement with previous studies[21], co-expression of $MenAT_3$ resulted in an increase in $M_r$ of 79 Da for $MenT_3$, corresponding to the addition of a phosphate (Fig. 1c; Supplementary Fig. 1C). In contrast, co-expression of $MenAT_4$ failed to produce a change in mass compared to $MenT_4$ expressed alone (Fig. 1c; Supplementary Fig. 1D). Unexpectedly, co-expression of $MenAT_1$ also resulted in an increase in $M_r$ of 80 Da for $MenT_1$ (Fig. 1c; Supplementary Fig. 1B). This provided evidence that $MenT_1$ was phosphorylated when co-expressed with $MenA_1$. Previous structural characterization of the $MenA_1$:$MenT_1$ complex likely failed to detect a phosphate within the toxin structures as both $MenA_1$ and $MenT_1$ were expressed independently and then co-incubated in vitro in the absence of phosphodonors[20]. This finding is supported by a recent phosphoproteomics study that identified both $MenT_1$ and $MenT_3$ phosphopeptides in *M. tuberculosis*[27]. Interestingly, and again in-line with our findings, no phosphopeptides with convincing probability scores were identified for $MenT_4$ (nor $MenT_2$)[27].

These data implied two post-translational modes of regulation for $MenAT_1$; sequestration and phosphorylation. We examined whether there was also any transcriptional regulation by cloning the 1000 bp region immediately upstream of the *menAT1* transcriptional start site into the promoterless *lacZ* fusion construct pJEM15[28], and quantified β-galactosidase activity in *Mycobacterium smegmatis* during co-induction of $MenA_1$, $MenT_1$, or both together (Fig. 1d; Supplementary Fig. 1F), using ParDE1 as a positive control for transcriptional repression[29]. Induction of either $MenA_1$, $MenT_1$, or both, failed to have an effect on β-galactosidase activity (Fig. 1d; Supplementary Fig. 1F), suggesting $MenAT_1$ cannot transcriptionally autoregulate.

### $MenA_1$ induces phosphorylation of $MenT_1$ in the presence of nucleotide substrates

To examine the requirements for $MenT_1$ phosphorylation we co-incubated $MenT_1$ in the absence or presence of combinations of $MenA_1$, $MgCl_2$, or CTP, the preferred NTase substrate of $MenT_1$[20]. Samples were then analysed by Phos-Tag SDS-PAGE and densitometry (Fig. 2a, b), alongside $ES^+$-ToF mass spectrometry (Supplementary Fig. 2). Samples from co-expression in vivo yielded a heterogenous toxin population of approximately equal abundances of phosphorylated and non-phosphorylated $MenT_1$ (Fig. 2a, b). In comparison, in vitro co-incubation of $MenT_1$ in the presence of $MenA_1$, $MgCl_2$ and CTP reproducibly generated a majority of phosphorylated $MenT_1$ (now referred to as $MenT_1$-p) (Fig. 2a, b). In the absence of magnesium there was less $MenT_1$-p generated, and the absence of either CTP or $MenA_1$ prevented production of $MenT_1$-p (Fig. 2a, b; Supplementary Fig. 2A–D). $ES^+$-ToF mass spectrometry confirmed that, as with CTP, $MenT_1$ was also phosphorylated when incubated with either ATP, GTP, or UTP in the presence of $MenA_1$ and $MgCl_2$ (Supplementary Fig. 2E–H), in agreement with previous in vitro activity assays demonstrating tRNA modification when incubated with each NTP[20]. Having identified increased phosphorylation activity in the presence of $MgCl_2$, we sought to establish whether magnesium was essential for $MenT_1$ phosphorylation. $MenT_1$ and $MenA_1$ were first pre-incubated with 5 mM EDTA for 1 h to facilitate chelation of protein-bound metals prior to overnight dialysis. Supplementation of protein mixtures with 1 mM CTP alone failed to induce a change in the mass of $MenT_1$ (Supplementary Fig. 2I). In contrast, supplementation with 1 mM CTP and 10 mM $MgCl_2$ produced an increase in mass of 81 Da, corresponding to the addition of a phosphate (Supplementary Fig. 2J), and thus confirming magnesium is essential for phosphorylation activity[29]. Next, we assessed the impact of mutations to highly conserved NTase fold residues T39, D41, K137,

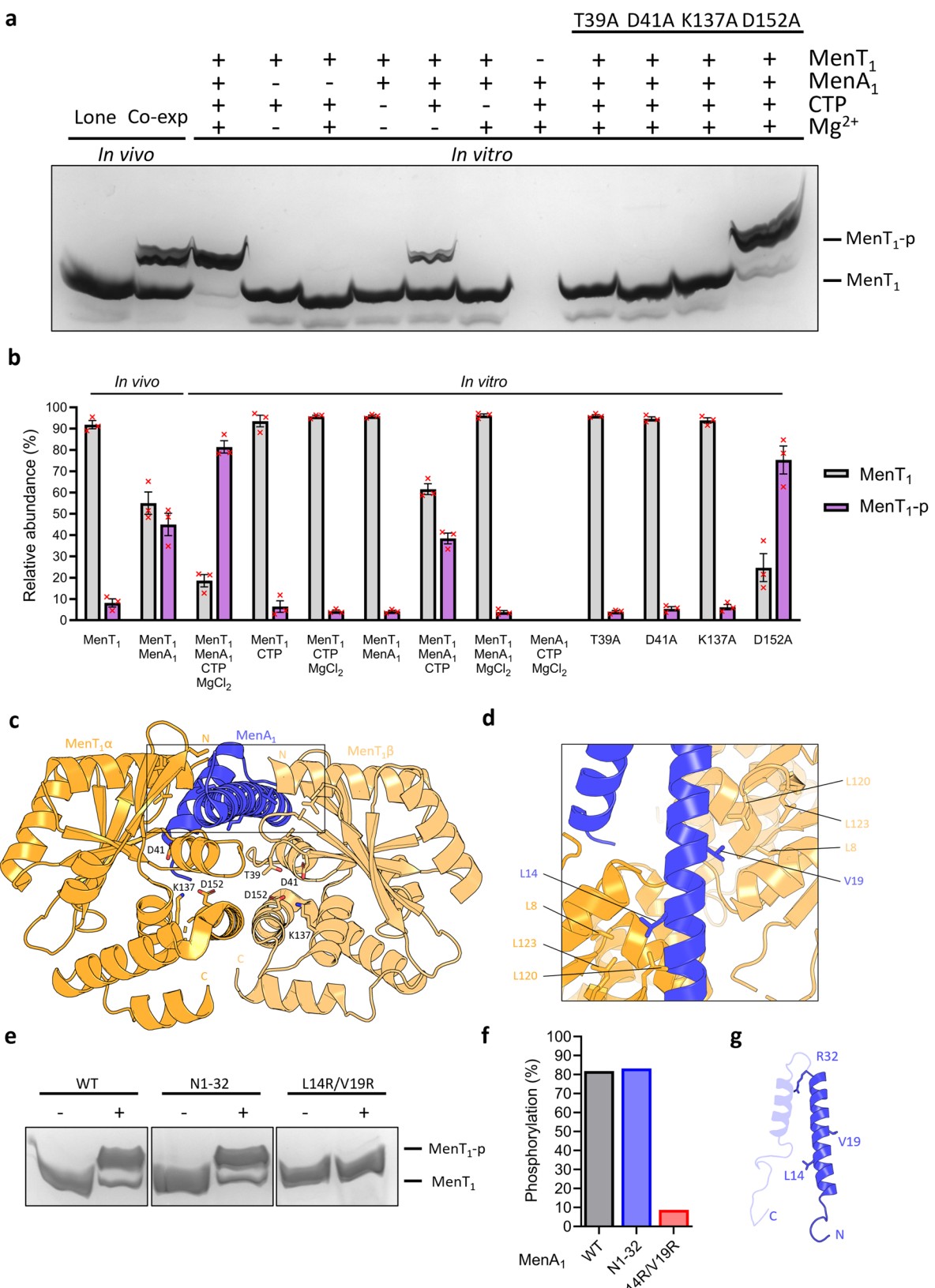

and D152 on phosphorylation activity, several of which were previously shown to be essential to NTase activity[8,20] (Fig. 2a, b; Supplementary Fig. 3). Toxin mutants T39A, D41A and K137A abolished phosphorylation, whereas D152A retained similar levels of phosphorylation activity to MenT$_1$ wild type (WT), but only with CTP, and to low levels with ATP (Fig. 2a–c; Supplementary Fig. 3A–C).

Next, we examined the role of MenA$_1$ in phosphorylation. Based on the MenAT$_1$ structure (PDB 8AN5; Fig. 2c, d) and previous activity assays[20], we tested a double mutant, L14R/V19R, and a truncation mutant encoding only the N-terminal α-helix (residues 1-32 (N1-32)), for their ability to induce in vitro phosphorylation of MenT$_1$ (Fig. 2e–g; Supplementary Fig. 4A–D). We previously reported that MenA$_1$ L14R/

**Fig. 2 | MenA$_1$ induces phosphorylation of MenT$_1$, resulting in a heterogenous toxin population. a** Phos-Tag SDS-PAGE of purified MenT$_1$ samples either expressed in the absence and presence of MenA$_1$, or co-incubated in the absence or presence of combinations of MenA$_1$, MgCl$_2$, and CTP. **b** Densitometric analysis of bands visualized by Phos-Tag SDS-PAGE. Data are representative of three independent biological replicates and bars represent the mean +/- SEM. **c** Crystal structure of the MenA$_1$:MenT$_1$ complex (PDB 8AN5) shown as a cartoon and coloured orange (MenT$_1$α), blue (MenA$_1$), and light orange (MenT$_1$β). MenT$_1$ conserved active site residues are shown as sticks for reference. **d** Close-up view of the boxed region in (**c**), rotated to display MenA$_1$ vertically, with residues partaking in hydrophobic interactions shown for reference. **e** Phos-Tag SDS-PAGE of purified MenT$_1$ incubated with MgCl$_2$ and CTP in the absence or presence of either wild-type MenA$_1$, or N1-32 or L14R/V19R mutants. **f** Densitometric analysis of (**e**) reveals phosphorylation activity is localized to the MenA$_1$ N-terminal α-helix, with L14R/V19R mutations inhibiting phosphorylation. Data are representative of two independent biological replicates. **g** Cartoon of MenA$_1$ with residues of interest shown as sticks. Source data are provided as a Source Data file.

V19R was unable to rescue MenT$_1$ toxicity in *M. smegmatis*[20], likely due to a loss of hydrophobic interactions between either residue and L8, L120, and L123 of MenT$_1$α/β (Fig. 2d). MenA$_1$ L14R/V19R caused reduced levels of MenT$_1$ phosphorylation compared to MenA$_1$ WT (Fig. 2e, f; Supplementary Fig. 4B and C). MenA$_1$ N1-32 lacks the C-terminal α-helix and disordered tail (Fig. 2g), but this did not impact phosphorylation of MenT$_1$ (Fig. 2e, f; Supplementary Fig. 4D). Based on these data, the stimulatory effect of MenA$_1$ on phosphorylation appears localized to the N-terminal α-helix.

Finally, we performed overnight co-incubations of MenA$_1$ with each nucleotide and analysed the resultant mixtures by thermal shift and ES$^+$-ToF mass spectrometry to confirm that MenA$_1$ does not directly bind nucleotides, and is itself not prone to phosphorylation (Supplementary Fig. 4E–H). When co-incubated with NTPs and MgCl$_2$ either in the absence of presence of the toxin (Supplementary Fig. 4F, G), no additional peaks were observed in the mass spectrum of MenA$_1$. Similarly, no significant changes to thermal stability were observed in the presence of nucleotides (Supplementary Fig. 4H). Collectively, these results demonstrate that MenA$_1$ antitoxin is required for phosphorylation of MenT$_1$ but is not itself phosphorylated.

## MenT$_1$ phosphorylation occurs at T39 and reduces the net positive charge within the conserved NTase catalytic core

We analysed the MenAT$_1$ co-expression sample by Liquid Chromatography Tandem Mass Spectrometry (LC-MS/MS), which confirmed that T39 is the site of MenT$_1$ phosphorylation (Supplementary Fig. 5A). No phosphorylation could be detected when MenT$_1$ T39A was co-expressed alongside MenA$_1$ (Supplementary Fig. 5B, C). A large-scale in vitro co-purification protocol was then devised in order to circumvent limitations in attaining a homogeneous phosphorylated toxin population following MenAT$_1$ co-expression. Utilizing this in vitro protocol we obtained sufficiently high yields of MenT$_1$-p for subsequent biochemical and crystallographic studies (Supplementary Fig. 5D). We solved the X-ray crystallographic structure of MenT$_1$-p to 2.80 Å (PDB 8RR5; Fig. 3a; Table 1). The MenT$_1$-p structure shows clear density for a phosphothreonine (TPO) at position T39 (Fig. 3b; Supplementary Fig. 5E). MenT$_1$-p was solved as a single toxin protomer with two gaps in external flexible loops spanning residues 87–94 and 142–150, the latter of which is a resolved loop in the crystal structure of non-phosphorylated MenT$_1$ (PDB 8AN4)[20]. Alignment of each structure via C-alpha atoms only returned an RMSD of 0.410 Å (160 atoms), indicating a near identical topology of core domains and folds (Fig. 3c). In contrast, aligning the C-alpha atoms of the respective T39 loops returned an RMSD of 0.778 Å (7 atoms), indicating a larger shift between these specific loops. Close-up views revealed further changes caused by the presence of the phosphothreonine, such as steric occlusion of the active site (Fig. 3d). Structural alignments also revealed that residues R37 and F38 are both ejected from the toxin active site following phosphorylation (Fig. 3d). Equivalent residues of R37 in structural homologues of MenT$_1$ have been shown to be essential to NTase activity[8] or are known to form hydrogen bonding interactions with incoming nucleotides[4], whilst F38 is involved in MenT$_1$ neutralization by MenA$_1$[20]. To predict the effect of phosphorylation on the surface electrostatics of MenT$_1$, we employed the PyMol APBS plugin to visualize differences in charge between MenT$_1$ and MenT$_1$-p. Overall, charge density appears similar between both structures, with enhanced electropositive charge localized to the N- and C-termini of MenT$_1$-p compared to MenT$_1$ (Fig. 3e). However, phosphorylation greatly reduces the net positive charge within the MenT$_1$ active site (Fig. 3e). Together, the movement of conserved residues away from the NTase core and the change in local charge provide clues as to how phosphorylation will have an impact both on MenT$_1$ toxicity and potentially its ability to bind MenA$_1$.

## MenA$_1$ antitoxin alters MenT$_1$ nucleotide specificity for NTase and phosphorylation activities

Next, thermal shift assays were used to detect altered stability of ligand-bound complexes and establish nucleotide preferences for MenT$_1$ NTase activity and phosphorylation. When incubated with each NTP and MgCl$_2$, then analysed by thermal shift, MenT$_1$ stabilization is significantly higher when incubated with CTP compared to other nucleotides ($p = 0.0004$, one-way ANOVA), reflected by a mean increase in melting temperature ($T_m$) of 3.8 °C, some 1.4 °C higher than when incubated with either ATP, GTP, or UTP (Fig. 4a–e). This matches the previously observed preference for CTP as a substrate for tRNA modification[20]. However, in the presence of 10 μM MenA$_1$, no significant differences between mean $\Delta T_m$ values were observed when incubated with any of the four nucleotides ($p = 0.323$, one-way ANOVA) (Fig. 4f). We subsequently performed Phos-Tag SDS-PAGE using a titration of each NTP in phosphorylation reactions to ascertain whether comparable thermal stabilization between nucleotides correlated to similar phosphorylation activities (Supplementary Fig. 6A). In agreement with thermal shift assays, each nucleotide substrate generated comparable levels of MenT$_1$-p across the range of tested concentrations (Supplementary Fig. 6A, B). This indicated that, unlike for toxic NTase activity, MenT$_1$ shows no NTP preference for phosphorylation. We then assayed MenT$_1$-p by thermal shift analysis to assess the impact of NTPs on already phosphorylated toxin. When incubated with each NTP in the absence or presence of antitoxin, MenT$_1$-p stability was greatly reduced, and in particular for ATP (Supplementary Fig. 6C, D). This indicates that the phosphate, in the presence of NTPs and with or without antitoxin, caused thermal destabilization of the toxin (Supplementary Fig. 6C, D).

Thermal shift assays were then repeated with toxin mutants T39A, D41A, K137A, and D152A in the presence of MgCl$_2$ and each NTP, either in the absence or presence of MenA$_1$ (Supplementary Fig. 6E). In agreement with Phos-Tag SDS-PAGE and ES$^+$-ToF Mass Spectrometry (Fig. 2a, b; Supplementary Fig. 3C), D152A retained the ability to bind to each NTP in the presence of MenA$_1$, with markedly higher stabilization when incubated with ATP or CTP (Supplementary Fig. 6E). Whilst T39A was not phosphorylated (Fig. 2a, b), it retained the ability to bind to each NTP in the presence of antitoxin, and so T39 is not involved in substrate binding. Conversely, all four NTPs failed to induce a change in melting temperature when incubated with D41A and K137A, suggestive of a loss of substrate binding (Supplementary Fig. 6E). To model substrate interactions we performed molecular docking of each NTP and Mg$^{2+}$ to the MenT$_1$ active site (Supplementary Fig. 7A). Best-scored docking poses showed that D41 co-ordinates Mg$^{2+}$, which in turn interacts with NTPs, whilst K137 directly interacts with the terminal γ-phosphate (Supplementary Fig. 7B). Furthermore, nucleotide

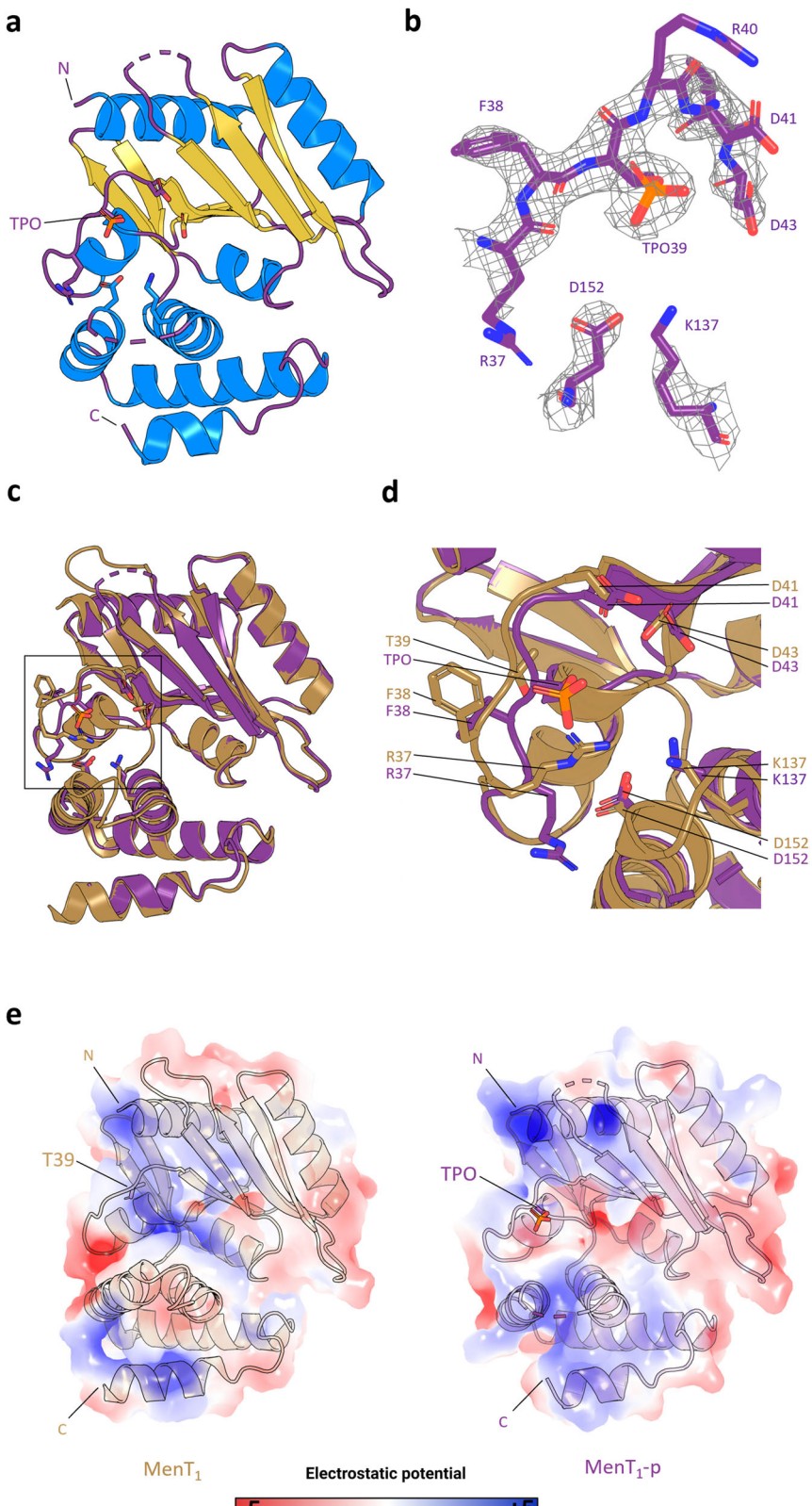

**Fig. 3 | MenT₁ T39 phosphorylation reduces the net positive charge within the NTase active site. a** Crystal structure of monomeric MenT₁-p (PDB 8RR5) shown as a cartoon and coloured by secondary structure elements. N and C termini are indicated. **b** 2F₀-F𝒸 electron density map of phosphorylated MenT₁ during structural refinement. **c** Structural alignment of MenT₁ (sand) and MenT₁-p (purple) protein backbones, RMSD 0.410 Å across 160 atoms. **d** Close-up view of the boxed region of (**c**), rotated 45 degrees around the *z*-axis. Residues of interest are shown as sticks coloured red for oxygen, blue for nitrogen, and orange for phosphorus. **e** Surface electrostatics of MenT₁ and MenT₁-p, viewed as in (**a**), depicting electrostatic potential from −5 $kBTe^{-1}$ (red) to +5 $kBTe^{-1}$ (blue), where e is the electron, $T$ is temperature and $k$B is the Boltzmann constant. Electrostatics were generated using default settings for the APBS plugin (PyMol).

**Table 1 | Data collection and refinement statistics**

| | MenT$_1$-p | MenT$_3$ |
|---|---|---|
| PDB ID code | 8RR5 | 8RR6 |
| Data collection | | |
| Space group | I 4 2 2 | P 32 2 1 |
| Cell dimensions | | |
| a, b, c (Å) | 127.00 127.00 68.78 | 95.27 95.27 69.03 |
| α, β, γ (°) | 90 90 90 | 90 90 120 |
| Resolution (Å) | 60.48–2.80 (2.95–2.80) | 41.25–1.78 (1.85–1.78) |
| $R_{merge}$ | 0.014 (0.074) | 0.0966 (5.678) |
| $R_{meas}$ | 0.020 (0.105) | 0.0992 (5.822) |
| I / σI | 27.4 (3.5) | 12.64 (0.27) |
| Completeness (%) | 100.00 (100.00) | 97.35 (75.14) |
| Redundancy | 1.8 (1.9) | 19.9 (20.0) |
| Refinement | | |
| Resolution (Å) | 2.80 | 1.78 |
| No. reflections | 13,277 (1956) | 692,864 (68,859) |
| Unique reflections | 7214 (1035) | 34,876 (3443) |
| $R_{work}$ | 0.1968 (0.4049) | 0.2065 (0.4160) |
| $R_{free}$ | 0.2394 (0.4144) | 0.2227 (0.4384) |
| No. atoms | | |
| Protein | 1357 | 2209 |
| Ligand/ion | 0 | 0 |
| Water | 4 | 79 |
| B-factors | 77.14 | 60.28 |
| Protein | 77.21 | 60.54 |
| Ligand/ion | – | – |
| Water | 54.60 | 53.22 |
| R.m.s. deviations | | |
| Bond lengths (Å) | 0.009 | 0.011 |
| Bond angles (°) | 1.71 | 1.46 |

One crystal per structure. Values in parentheses are for highest-resolution shell.

recognition appeared to be primarily localized to the triphosphate tail, as reported for other NTases[30], with few detectable protein-ligand interactions between any of the structurally divergent nucleotide bases and MenT$_1$ (Supplementary Fig. 7B). Each nucleotide displayed similar overall poses, indicating conserved binding mechanisms (Supplementary Fig. 7A, B). This matches an observed lack of nucleotide specificity exhibited by other NTases in the absence of tRNA targets[3] and presents a structural explanation for the ability of MenT$_1$ to utilize ATP, GTP, and UTP alongside CTP as substrates for phosphorylation[20]. Furthermore, the proximity of each NTP to T39 following molecular docking suggests that the observed destabilization of MenT$_1$-p when co-incubated with nucleotides is likely a result of electrostatic repulsion between the phosphothreonine of MenT$_1$-p and the negatively charged triphosphate backbone (Supplementary Figs. 6C, D; 7A, B).

Following docking, we speculated whether nucleotide di- and mono-phosphates may also function as viable phosphodonors. We repeated phosphorylation assays using derivatives of ATP, namely AMP-PNP, ADP, and AMP, to establish which phosphate(s) could be used to generate the phosphothreonine. Co-incubation of MenT$_1$, MenA$_1$ and MgCl$_2$ with ADP, but not AMP or AMP-PNP, supported phosphorylation (Supplementary Fig. 7C), confirming that both the β- and γ- phosphates can be utilized as donors.

### Phosphorylation of MenT$_1$ inhibits NTase activity and prevents de novo heterotrimeric complex formation

Based on our solved crystal structure of MenT$_1$-p and its reduced thermal stability when co-incubated with nucleotide substrates

(Supplementary Fig. 6C, D), we hypothesized that phosphorylation within the MenT$_1$ active site would reduce NTase activity. Accordingly, when MenT$_1$ and MenT$_1$-p were incubated with *E. coli* tRNAs from the cell-free PURExpress translation system, we found that MenT$_1$-p was unable to inhibit GFP protein synthesis (Fig. 5a). Similarly, co-incubation of MenT$_1$-p with radioactively labelled tRNA Gly-3 in the presence of [α$^{32}$P]-CTP failed to result in modification of tRNA compared to when using MenT$_1$ (Fig. 5b). Phosphorylation had the same impact on NTase activity as inactivating mutations D41A and K137A (Fig. 5b). The same experiment was performed using total RNA extracts from *M. smegmatis* (Fig. 5c). Again, MenT$_1$-p was inactive, as were mutants D41A, K137A and D152A (Fig. 5c). Interestingly, MenT$_1$ T39A was still active but at a lower level, suggesting T39 is important for full activity, but is not required for toxic NTase activity (Fig. 5c). This conclusion is supported by data showing nucleotide binding by T39A (Supplementary Fig. 6E) and continued in vivo toxicity of the T39A mutant[20]. Next, we aimed to separate the antitoxic impacts of sequestration and phosphorylation by monitoring NTase activity in the presence of MenA$_1$, using either MenT$_1$ WT or non-phosphorylatable T39A. Incubation of MenT$_1$ T39A with MenA$_1$ reduced in vitro tRNA modification levels, but higher concentrations of MenA$_1$ were required to achieve comparable levels of MenT$_1$ T39A inhibition to those of MenT$_1$ WT (Fig. 5d). This result shows that T39 is not essential for toxicity or antitoxicity, and more importantly indicates that the two modes of antitoxicity, sequestration and phosphorylation, appear to be additive. To examine the requirement of T39 for toxic activity, we generated a second mutant, T39C, which more closely resembles the native threonine, and tested its ability to inhibit *M. smegmatis* growth. Both MenT$_1$ T39A and T39C mutants remained toxic when expressed in *M. smegmatis*, though growth inhibition appeared weaker than observed for MenT$_1$ WT (Fig. 5e). MenT$_1$ T39A and T39C toxicity could still be abolished when the mutants were expressed in the presence of MenA$_1$ (Fig. 5e). T39 is therefore important for full toxic activity but not essential, and antitoxicity can still occur with an abundance of MenA$_1$ to provide sequestration, even in the absence of a target for phosphorylation.

Next, we examined MenAT$_1$ complex formation in the absence and presence of phosphorylation. We co-incubated MenT$_1$ WT or T39A with MenA$_1$ in the absence of NTPs and analysed the resultant mixtures by analytical SEC (Fig. 5f–h, Supplementary Fig. 8). Co-incubation of either sample with MenA$_1$ resulted in peaks matching the predicted elution volume of the MenA$_1$:MenT$_1$ heterotrimer (Fig. 5f, g). Overlaying either peak onto the elution profiles of known molecular weight calibrants indicates both species are ~40 kDa (Supplementary Fig. 8A–D). Comparison of the observed Stokes radii ($R_{st}$) for each against the calculated $R_{st}$ of the MenA$_1$:MenT$_1$ heterotrimer returned ratios of 0.97 and 0.94 for MenT$_1$ and MenT$_1$ T39A respectively (Supplementary Fig. 8E, F), validating that each peak corresponds to the heterotrimeric MenA$_1$:MenT$_1$ complex. In contrast, co-incubation of MenA$_1$ and MenT$_1$-p resulted in a far less pronounced increase in molecular weight than had been observed for MenT$_1$ WT or T39A (Fig. 5h; Supplementary Fig. 8A). The observed $R_{st}$ value of this peak was 23.68 Å, which when compared against the calculated $R_{st}$ of the MenA$_1$:MenT$_1$ heterotrimer returned an observed/calculated ratio of 0.79, suggesting this species is unlikely to be the heterotrimeric complex (Supplementary Fig. 8G). Nevertheless, the observed peak indicated that a larger species had indeed been formed as a result of MenA$_1$-MenT$_1$-p co-incubation. By using AlphaFold to generate a predictive model of heterodimeric MenA$_1$:MenT$_1$ (predicted template modelling (pTM) score 0.89), we calculated an approximate $R_{st}$ value to be correlated against the observed $R_{st}$ of the unknown species. When compared against the calculated $R_{st}$ of this model, the observed $R_{st}$ of the unknown species almost perfectly matches that of the hypothetical heterodimer, returning an observed/calculated $R_{st}$ ratio of 0.97 (Supplementary

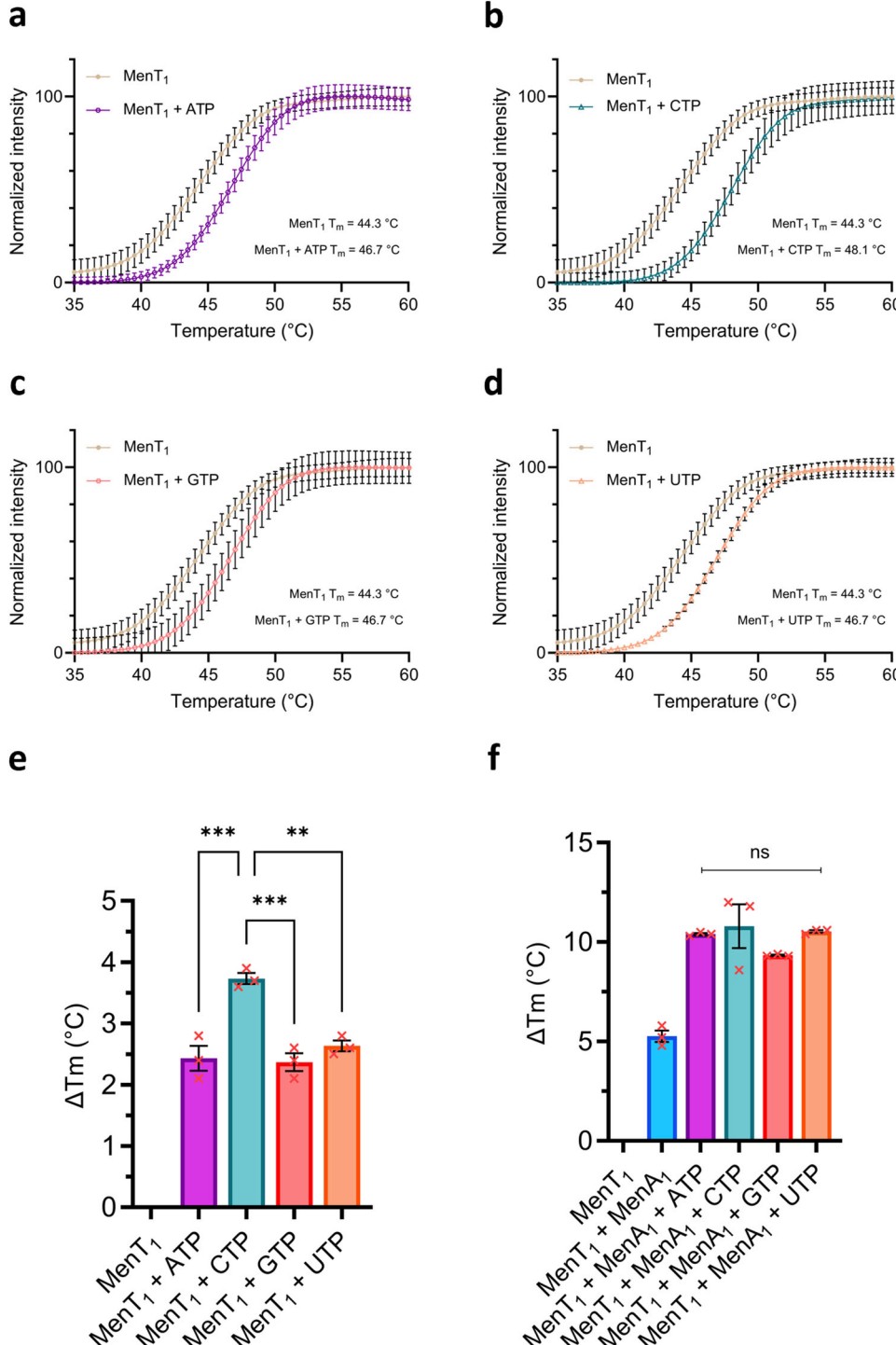

**Fig. 4 | MenA₁ abolishes MenT₁ nucleotide specificity. a–d** Thermal shift isotherms of MenT₁ incubated with MgCl₂ in the absence and presence of either ATP (**a**), CTP (**b**), GTP (**c**), or UTP (**d**). Isotherms are normalized between minima and maxima for presentation and comparison, cropped to the appropriate scale. **e, f** Mean changes in melting temperature following overnight incubation of MenT₁ with MgCl₂ and either ATP, CTP, GTP, or UTP, in the absence (**e**) and presence (**f**) of MenA₁ (one-way ANOVA; (**e**) *** $p = 0.0004$; (**f**) $p = 0.323$). Plotted data represent the mean +/- SEM (3 replicates). Source data are provided as a Source Data file.

Fig. 8G). These data suggest that MenT₁-p can form a stable heterodimer, but not a heterotrimeric complex.

### MenA₁ N-terminal α-helix triggers sequential MenT₁ autophosphorylation

All-atom molecular dynamics (MD) simulations were conducted to investigate potential mechanisms of MenT₁ phosphorylation. MD simulations placing ATP and a single Mg²⁺ ion within the native MenT₁ monomeric structure (PDB 8AN4) revealed that the majority of binding-site interactions are formed between the conserved NTase core and nucleotide phosphoryl oxygens (Fig. 6a, b), as has been reported for other NTases[4,31]. This result also matches poses previously obtained by docking (Supplementary Fig. 7A and B). Within MenT₁ apo, the γ-phosphate of ATP is predicted to interact with R40, R84, and R146, whilst the conserved D₄₁xD₄₃ motif co-ordinates Mg²⁺, which in turn anchors the α- and β-phosphates and locks ATP in a kinked

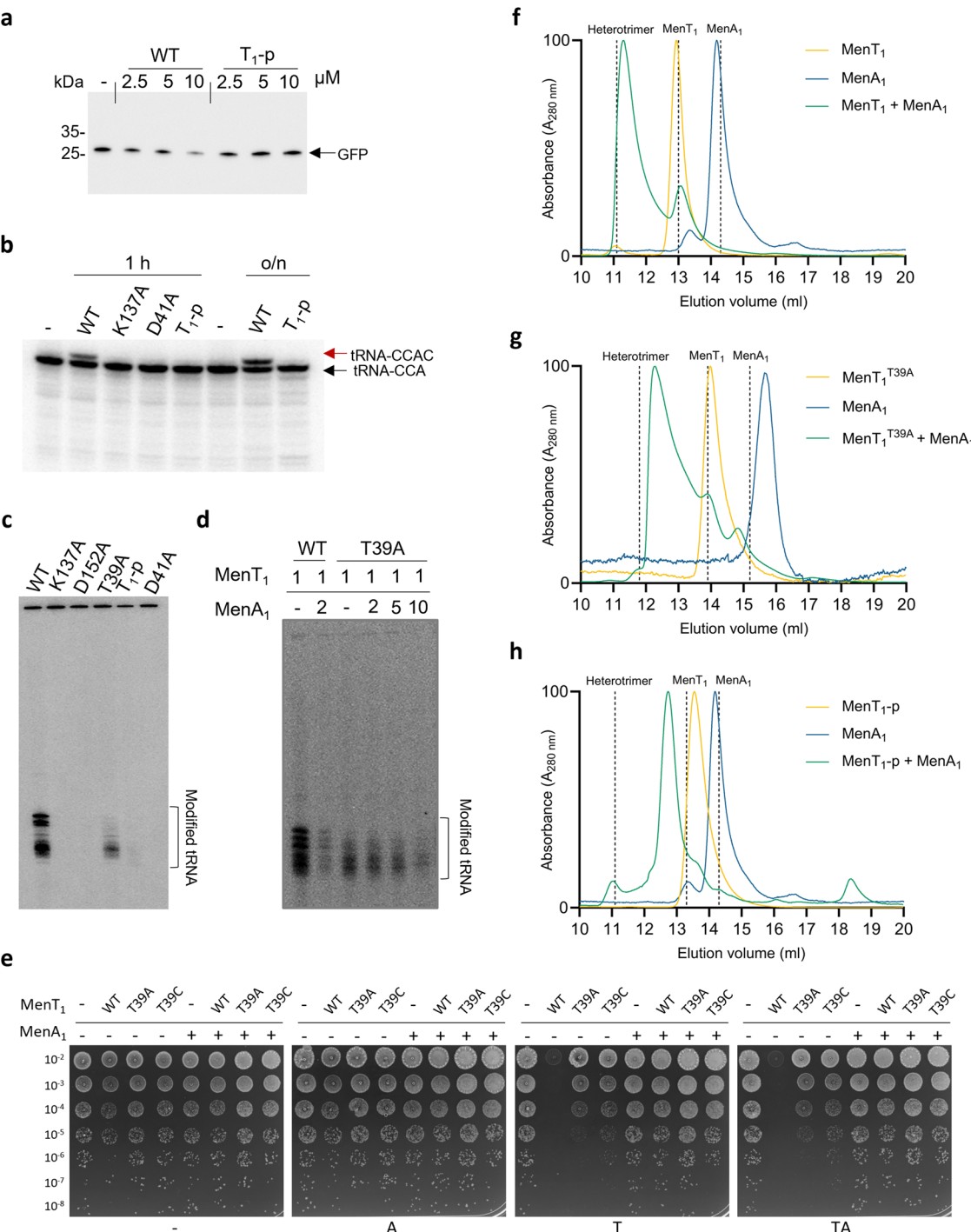

**Fig. 5 | Phosphorylation of MenT1 inhibits NTase activity. a** Total tRNA from *E. coli* was pre-incubated with MenT1 in vitro and subsequently used in a cell-free translation assay. Samples were separated on 4–20% SDS-PAGE gels and expression levels of GFP model substrate were determined using anti-GFP antibody. **b** [α-$^{32}$P]-CTP labelled *M. tuberculosis* tRNA Gly-3 was incubated with MenT1 or its variants (5 µM) for 4 h at 37 °C in the presence of unlabelled CTP. Red arrows indicate the presence of cytidine extension. **c, d** Total RNA from *M. smegmatis* was incubated with MenT1 or its variants (5 µM) in the presence of [α-$^{32}$P]-CTP at 37 °C for 2 h. **e** Toxicity/antitoxicity assays were performed in *M. smegmatis* to study the importance of MenT1 T39 in toxicity/antitoxicity. Co-transformants of *M. smegmatis* containing pGMC -vector (-), -MenT1 WT, or -MenT1 T39A and T39C variants, and pLAM -vector (-) or -MenA1 WT were serially diluted and spotted on LB agar plates in the presence or absence of toxin and antitoxin inducers (100 ng ml$^{-1}$ ATc or 0.2% Ace, respectively). Plates were incubated for 3 days at 37 °C. "A" = antitoxin induced, "T" = toxin induced, "TA" = antitoxin + toxin induced. **f–h** Overlaid analytical SEC traces from a Superdex™ 75 increase 10/300 GL SEC column corresponding to either MenT1 (**f**), MenT1 T39A (**g**), or MenT1-p (**h**) incubated in the absence and presence of MenA1, confirming heterotrimeric complex formation blocks MenT1 T39A toxicity. Chromatograms are normalized between 0 and 100 for presentation and comparison, cropped to the appropriate scale. Vertical dashed lines display the expected elution volume of respective samples based on calculated Stokes Radii. Data are representative of three independent biological replicates. Source data are provided as a Source Data file.

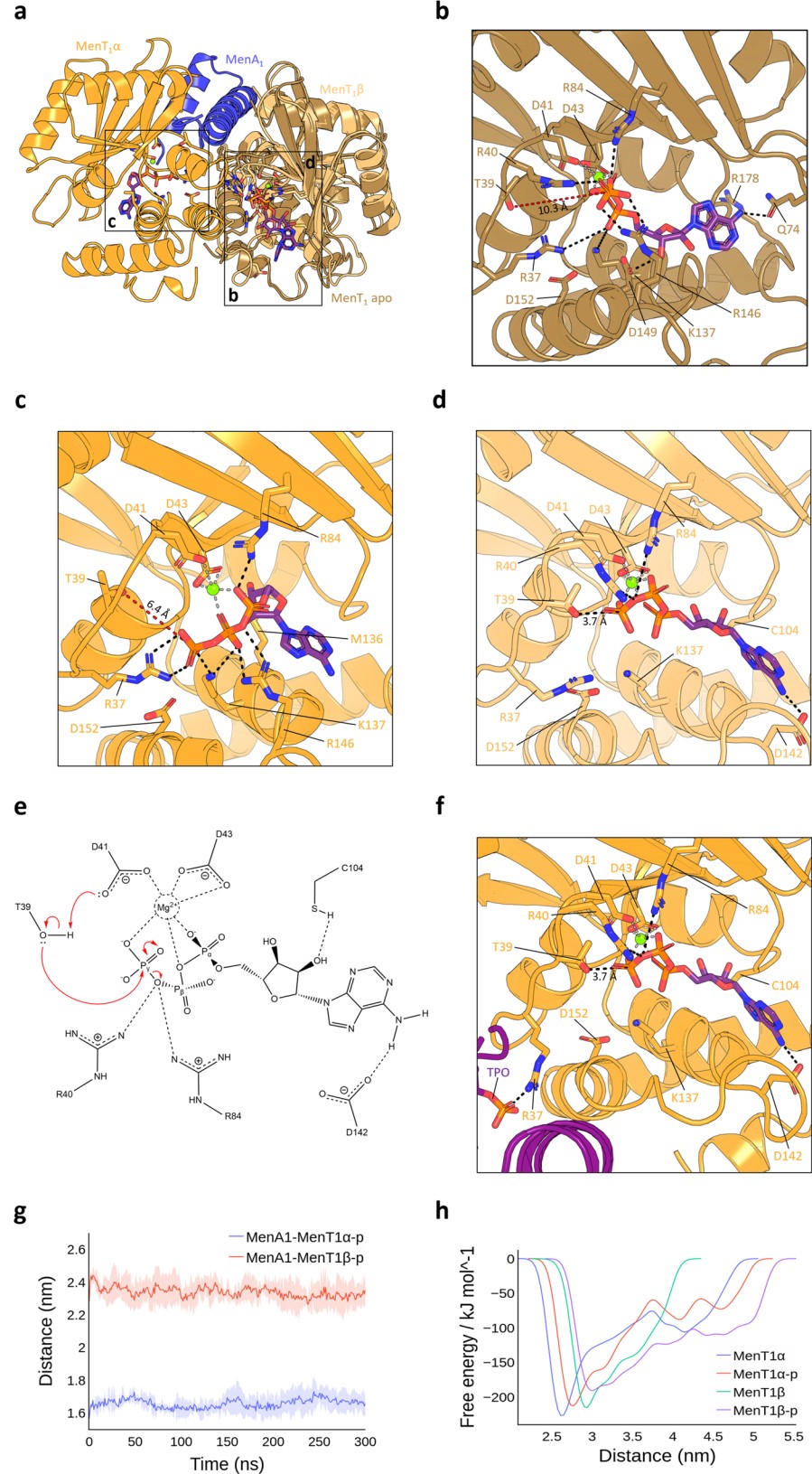

orientation (Fig. 6b). Together with R37, K137 stabilizes the ATP α-phosphate oxygens (Fig. 6b), supporting the conclusion that the inability of K137A to phosphorylate is a result of a loss of nucleotide binding (Fig. 2a, b; Supplementary Fig. 3B). D149 is predicted to interact with the 3′ OH of the ribose sugar, whilst Q74 forms the only interaction with the adenine base (Fig. 6b). In this model, the γ-

phosphate of ATP lies 10.3 Å from T39, presenting a possible rationale for the observed lack of toxin phosphorylation in the absence of antitoxin.

MD simulations of the MenA$_1$:MenT$_1$ heterotrimer (PDB 8AN5) bound to ATP revealed asymmetry in the ATP positioning and protein-ligand interactions. In MenT$_1$α, similar surface contacts with

**Fig. 6 | MenA₁ induces auto-phosphorylation of MenT₁. a** Structural overlay of MenT₁ and MenA₁:MenT₁ bound to ATP following MD simulations, shown as cartoons coloured sand (MenT₁ apo), orange (MenT₁α), blue (MenA₁), and light orange (MenT₁β). **b–d** Close-up views of the boxed regions in (**a**) depicting distances between the γ-phosphate of ATP and T39 in MenT₁ apo (**b**), and either MenT₁α (**c**) or MenT₁β (**d**) of the MenA₁:MenT₁ complex. Residues of interest are shown as sticks coloured red for oxygen, blue for nitrogen, and orange for phosphorous, with dashed lines shown to depict protein-ligand interactions. **e** Proposed mechanism of toxin auto-phosphorylation. Interacting residues are displayed with hydrogen bonds shown as horizontal dashed lines. **f** Close-up view of ATP bound to the MenT₁α active site following MD simulations of MenT₁β phosphorylation within the heterotrimer. MenT₁β-p is shown as a purple cartoon. **g** Average distance plots depicting distances between phosphorylated MenT₁α/MenT₁β and MenA₁ following MD simulations. **h** Free-energy profiles depicting average distances between toxin protomers and MenA₁ in complex during MD simulations. Source data are provided as a Source Data file.

ATP are observed as within monomeric MenT₁, and the ATP γ-phosphate remains at a significant distance from T39, although this is reduced to 6.4 Å in the MenA₁:MenT₁ heterotrimer compared to MenT₁ apo (Fig. 6b, c). In contrast, ATP bound to MenT₁β shows the distance between T39 and the ATP γ-phosphate to be reduced to 3.7 Å, with ATP adopting a more extended pose than in MenT₁ apo or MenT₁α (Fig. 6d). When bound to MenT₁β, D142 and C104 form hydrogen bonding interactions with the adenine N-6 and ribose 2′ OH, respectively, whilst R40 and R84 directly interact with the β,γ-bridging oxygen (Fig. 6d). We concluded that MenA₁ asymmetrical binding has forced movement of the MenT₁β T39 loop towards the bound ATP, facilitating auto-phosphorylation of MenT₁β. In this model, MenT₁β D41 also lies 3.7 Å from T39, which we propose facilitates proton extraction and activation of the nucleophilic hydroxyl sidechain of T39 (Fig. 6e), a trademark characteristic of conserved catalytic bases in NTases and kinases alike[31,32]. Activated T39 can then attack the γ-phosphate, allowing associative displacement of the phosphate through a remodelled trigonal bipyramidal transition state (Fig. 6e). Although MenA₁ binding would be sufficient for phosphorylation of MenT₁β, the distance of MenT₁α T39 from the ATP γ-phosphate led us to speculate that a second conformational change within the heterotrimer would be required to facilitate phosphorylation of MenT₁α. We subsequently simulated phosphorylation of MenT₁β and compared radius of gyration plots for non-phosphorylated (MenT₁α:MenA₁:MenT₁β) and mono-phosphorylated (MenT₁α:MenA₁:MenT₁β-p) trimers. The results of our extended simulations show that MenT₁α:MenA₁:MenT₁β exhibits a more compact structural arrangement than MenT₁α:MenA₁:MenT₁β-p, with phosphorylation increasing the overall flexibility of the complex (Supplementary Fig. 9A). No detrimental effects to complex stability were detected following phosphorylation of MenT₁β (Supplementary Fig. 9B). Comparison of average distances between either protomer in the non-phosphorylated complex reveals both reside at a similar mean distance from MenA₁ during the entirety of the simulation (Supplementary Fig. 9C). However, phosphorylation of MenT₁β results in a sharp decrease in the distance between MenA₁ and MenT₁α at ~75 ns (Supplementary Fig. 9C). The resulting pose indicates that the phosphothreonine of MenT₁β directly interacts with MenT₁α R37, thereby pivoting MenT₁α T39 to within 3.7 Å of the ATP γ-phosphate, the same distance observed as a starting point for the interaction of MenT₁β and ATP (Fig. 6d, f). Based on our extended simulations, R40, R84, C104, and D142 of MenT₁α are predicted to form identical protein-ligand interactions as had been observed for the predicted MenT₁β-ATP pose, indicating that phosphorylation of MenT₁β causes MenT₁α to become competent for auto-phosphorylation.

Overall, MD simulations modelled how within the unphosphorylated heterotrimer, MenA₁ induces the auto-phosphorylation of MenT₁ by promoting a conformational change that positions T39 of MenT₁β in close proximity to the γ-phosphate of ATP (Fig. 6a–d). The key region in MenA₁ facilitating movement is residues 1–32, in agreement with the ability of the α1 helix to induce MenT₁ phosphorylation (Fig. 2e, f) and inhibit toxicity in *M. smegmatis*[20]. Auto-phosphorylation of MenT₁β then promotes a second conformational change that enables auto-phosphorylation of MenT₁α, thereby rendering both bound toxins inert.

We then performed equilibrium MD simulations and well-tempered metadynamics using the dual-phosphorylated heterotrimer (MenT₁α-p:MenA₁:MenT₁β-p) to assess its stability, as in vitro co-incubations of MenA₁:MenT₁-p evidenced the existence of a phosphorylated heterodimer in solution, but failed to detect the presence of phosphorylated heterotrimer (Fig. 5h; Supplementary Fig. 8C). Calculations of the centroid distances between MenA₁ and each toxin protomer in the hypothetical MenT₁α-p:MenA₁:MenT₁β-p complex revealed that MenT₁α-p is, on average, 0.8 Å closer to MenA₁ than MenT₁β-p (Fig. 6g). Furthermore, in the non-phosphorylated trimer, MenT₁α displayed lower overall energy at the base of its free-energy valley and remained closer to MenA₁ at its free-energy minima compared to MenT₁β (Fig. 6h). However, as both phosphorylated protomers attempt to move away from MenA₁, the free energy curve of MenT₁α-p increases more steeply than MenT₁β-p, suggesting that interactions are inherently more stable between MenA₁ and MenT₁α-p (Fig. 6g). These observations are corroborated by protein interaction energy plots suggesting a tendency of the phosphorylated trimer to dissociate into a MenA₁:MenT₁α-p heterodimer and release MenT₁β-p, as evidenced by far weaker interactions between MenT₁β-p and MenA₁ (Supplementary Fig. 9D and Supplementary Movie 1). Collectively, the equilibrium MD and enhanced sampling simulations fit our experimental data and present a mechanism for sequential MenT₁ auto-phosphorylation induced by MenA₁, and production of a MenA₁:MenT₁-p heterodimer. The role of this phosphorylated heterodimer in cells remains unclear, however, as we failed to detect the presence of a heterodimeric species during SEC following MenAT₁ co-expression in vivo (Fig. 1b), this complex may be transient and not have a major role in regulating toxicity.

### A conserved mechanism of auto-phosphorylation for NTase regulation

MenA₁ and MenA₃ are unrelated (Fig. 1a), and have very different structures (MenA₁ PDB 8AN5; MenA₃ based on Alphafold models (pTM score 0.87) and homology to AbiEi PDB 6Y8Q). Like MenA₁, MenA₃ does not have notable kinase motifs, and interacts with MenT₃ during in vivo co-purification assays[6], though interactions are weaker than observed for MenAT₁ (Fig. 1b). We hypothesized that the proposed mechanism of antitoxin-dependent toxin auto-phosphorylation could be applied to MenT₃. We first sought to establish whether phosphorylation of MenT₃ S78 results in steric occlusion of the conserved NTase cavity, as had been observed for MenT₁ and MenT₁-p (Fig. 3b, c). We solved the crystal structure of non-phosphorylated MenT₃ WT to 1.78 Å (PDB 8RR6) (Fig. 7a; Table 1) and overlaid this structure against that of MenT₃-p (PDB 6Y5U). Structural alignment of respective protein backbones returned an RMSD of 0.144 Å (268 atoms), and there was some small movement of the S78 loop, leading to occlusion of the active site by the phosphoserine (SEP) (Fig. 7b), albeit to a lesser extent than we observed for MenT₁-p (Fig. 3c). Using the solved crystal structure of MenT₃ as a starting model, we performed MD simulations of ATP and Mg²⁺ binding in the absence and presence of MenA₃ to establish whether a conformational change to the S78-bearing loop ensued (Fig. 7c). In the absence of antitoxin, R76 and K79 stabilize the γ-phosphate of ATP, which lies 10.0 Å from the sidechain of S78 (Fig. 7d). This arrangement closely resembles the protein-ligand interactions observed for the MenT₁-ATP pose (Fig. 6b). H207 and K189 interact with the β-phosphate, whilst R205 directly interacts with

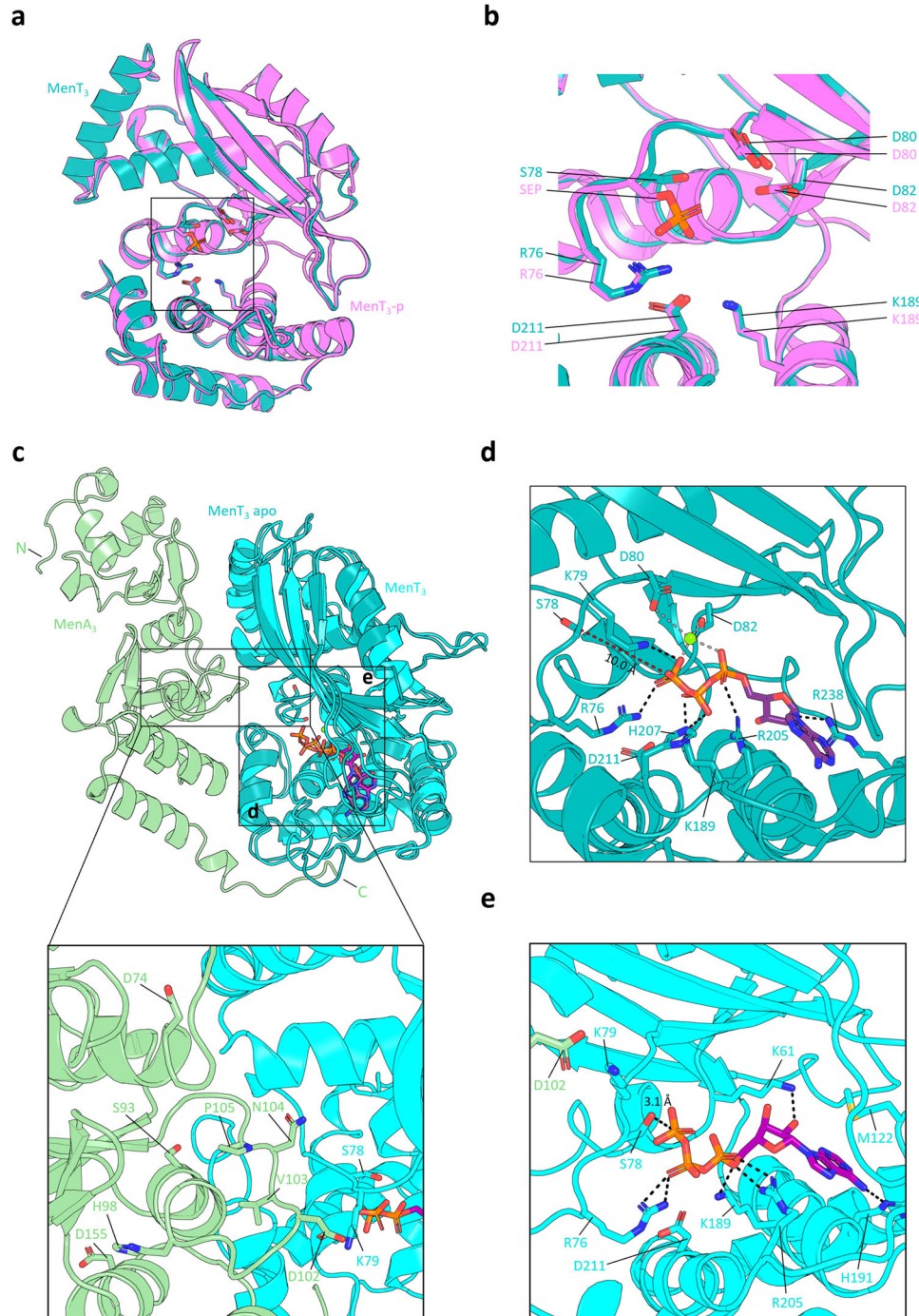

**Fig. 7 | Molecular dynamics predicts a conserved mechanism of MenT autophosphorylation. a** Structural overlay of MenT₃ (PDB 8RR6, teal) and MenT₃-p (PDB 6Y5U, violet), RMSD 0.158 Å across 1808 atoms. **b** Close-up view of the boxed region in (**a**). Residues of interest are shown as sticks red for oxygen, blue for nitrogen, and orange for phosphorus. **c** Structural overlay of MenT₃ and the hypothetical MenA₃:MenT₃ heterodimer bound to ATP following molecular dynamics simulations, shown as cartoons coloured teal (MenT₃ apo), cyan (MenT₃ complex) and green (MenA₃), with a zoomed and rotated view showing the predicted MenA₃:MenT₃ interaction interface. **d**, **e** Close-up views of the boxed regions in (**c**) depicting distances between the γ-phosphate of ATP and S78 in MenT₃ apo (**d**) and MenT₃ from the MenA₃:MenT₃ complex (**e**). Residues of interest are shown as sticks coloured red for oxygen, blue for nitrogen, and orange for phosphorous, with dashed lines shown to depict protein-ligand interactions.

the α-phosphate, and the only non-phosphate interaction is formed between the ribose 2′ OH and R238 (Fig. 7d). In agreement with our simulations, the recently solved crystal structure of CTP bound to MenT₃ (PDB 8XHR) revealed similar interactions exist between MenT₃ and the triphosphate tail region of CTP (Supplementary Fig. 9E–G)[33], with MenT₃ S78 some 8.2 Å from γ-phosphate of CTP (Supplementary Fig. 9E). Despite the overall high levels of similarity between ATP and

CTP binding poses (Supplementary Fig. 9G), there are notable differences between our predictive MenT₃-ATP pose and that of the MenT₃-CTP crystal structure. Specifically, hydrogen bonding and base-stacking interactions between MenT₃ P120, M122, and R238 and the cytidine base of CTP are absent in the MenT₃-ATP pose (Supplementary Fig. 9E, F). This lack of protein-nucleotide base interactions in the predictive ATP-bound model may explain why MenT₃ exhibits

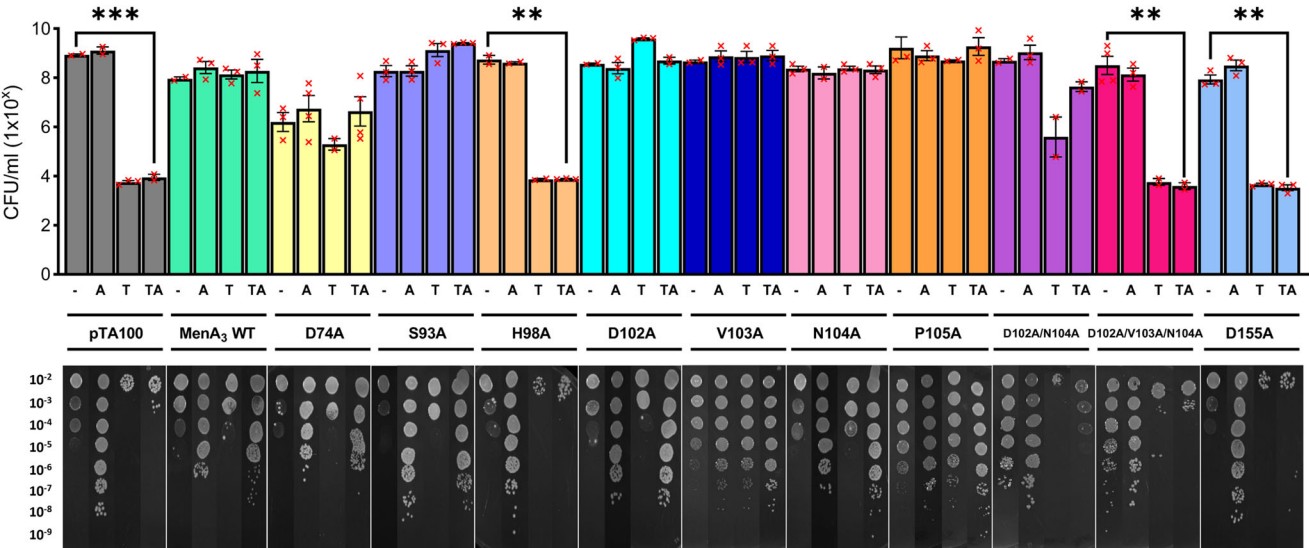

**Fig. 8 | MenA$_3$ α5-β4 loop residues are essential for activity.** Endpoint viable count antitoxicity assays of *E. coli* DH5α transformed with pPF657 (MenT$_3$) and either pTA100 empty vector (-), pPF656 (MenA$_3$ WT), or plasmids pTRB669-674 and pTRB717-720 (MenA$_3$ mutants). Overnight cultures were re-seeded into fresh LB supplemented with Ap, Sp and D-glu and grown to mid-log phase. Samples were serially diluted $10^{-2}$–$10^{-9}$ and spotted on M9A plates containing Ap and Sp, and with or without D-glu, L-ara and IPTG for repression of toxin expression, induction of toxin expression, and induction of antitoxin expression, respectively. Plates were incubated at 37 °C for 48 h, after which they were imaged and colonies counted to determine CFU/ml (one-way ANOVA, *** $p < 0.001$). "A" = antitoxin induced, "T" = toxin induced, "TA" = antitoxin + toxin induced. Plotted data represent the mean +/- SEM (3 replicates). Source data are provided as a Source Data file.

preference for CTP as an NTase substrate over other nucleotides. Interestingly, it was also proposed that the equivalent residue to R238 in MenT$_1$, R178, may be responsible for governing nucleotide specificity[33]. However, in our models and in the native MenT$_1$ crystal structure, R178 lies far displaced from the toxin active site (Fig. 6b), suggesting an alternative mechanism for specificity exists for MenT$_1$. When docked to the Alphafold-predicted MenA$_3$:MenT$_3$ heterodimer used to run the simulations (pTM score 0.84), the looped region of MenA$_3$ connecting α5-β4, spanning residues H98-I108, forces MenT$_3$ S78 into an inwards-facing conformation akin to binding of MenA$_1$ to MenT$_1$ (Fig. 7c). As a result, the overall distance between S78 and the terminal phosphate of ATP is reduced to 3.1 Å (Fig. 7e). In addition, R76 maintains contact with the β-phosphate, with R205 and K189 predicted to stabilize the α-phosphate (Fig. 7d, e). Hydrogen bonding is also observed between the adenine N-6 and the sidechain of the adjacent H191, and K61 binds the ribose 2′ OH. Finally, M122 stacks with the adenine base, similar to the MenT$_3$-CTP pose (Supplementary Fig. 9E).

To validate the role of the MenA$_3$ looped region in facilitating phosphorylation of MenT$_3$, we performed site-directed mutagenesis on a series of residues predicted to interface with the toxin proximal to its active site, then subsequently tested the ability of these antitoxin mutants to attenuate toxicity. Endpoint viable counts revealed that single H98A and D155A mutations inhibited antitoxicity (Fig. 8), in agreement with existing literature[21]. Single D74A, S93A, D102A, V103A, N104A, and P105A mutations failed to abolish antitoxic activity. The ability of the D74A, S93A and P105A mutants to still provide antitoxicity was surprising, as it was previously reported that phosphorylation activity was inhibited by the loss of these specific residues[21]. In contrast, double D102A/N104A mutations resulted in an intermediate phenotype whereby antitoxic activity appeared slightly diminished, though statistical analyses failed to demonstrate a significant difference relative to induction of MenT$_3$ alone (Fig. 8). In line with our predictive MD simulations, triple D102A/V103A/N104A mutations inhibited the ability of MenA$_3$ to counter MenT$_3$ toxicity, confirming that Alphafold-predicted interacting residues are indeed essential for MenA$_3$ activity. Whilst the specific role of these residues in mediating antitoxicity requires further exploration, our findings support the existence of a conserved mechanism of NTase regulation used by structurally divergent MenA$_1$ and MenA$_3$ antitoxins.

## Discussion

This study provides biochemical, structural, biophysical and computational characterization of an alternative mode of antitoxicity within TA systems. We predict that neutralization of MenT$_1$ toxicity can occur by auto-phosphorylation of the NTase active site, as a direct result of MenA$_1$ antitoxin binding. We then show this could be a general mechanism used to regulate NTases, as a second toxin, MenT$_3$, undergoes similar auto-phosphorylation despite fundamental differences in the acting antitoxin partner.

Unlike the 22.5 kDa MenA$_3$, MenA$_1$ is a mere 7.4 kDa and lacks the conserved winged helix-turn-helix DNA-binding and C-terminal antitoxicity domains. Both antitoxins lack conserved kinase domains or motifs[34], and based upon the ability of MenA$_1$ truncation mutants to phosphorylate MenT$_1$, we have demonstrated that minimal binding partners, even a single α-helix, can be sufficient to induce auto-phosphorylation (Fig. 2e, f). Subsequently, MD simulations independently generated the same mechanism for each TA pair, wherein antitoxin binding induces a substantial conformational change, moving the target phosphoacceptor towards the donor phosphate (Figs. 6 and 7). Mutagenesis studies of MenA$_3$ further support our model, with loss of D102/V103/N104 residues resulting in the inhibition of antitoxic activity, as all these residues are encoded on the looped region of MenA$_3$ that we propose binds and forces a conformational change within MenT$_3$[21] (Fig. 7c). In neither system are residues of the antitoxin actually within the ligand binding pocket and taking part in the proposed mechanism. We therefore conclude that regulation of MenT$_1$ and MenT$_3$ NTases occurs by antitoxin-induced toxin auto-phosphorylation.

Akin to protein kinases, the catalytic core of DUF1814 NTases houses a highly conserved aspartate involved in Mg$^{2+}$ chelation (DxD), and a catalytic lysine that directly interacts with the terminal phosphate of incoming nucleotides[1,32]. However, like MenA antitoxins, MenT toxins are devoid of kinase architecture[34], indicating that shared homology is restricted solely to metal co-ordination and substrate

binding/activation. Studies probing the mechanism of *M. tuberculosis* Protein Kinase B (PknB) auto-phosphorylation reported similar distances between the target phosphoacceptor and NTP γ-phosphate as those predicted by our docking models and MD simulations[35]. This demonstrates that the proposed conformational changes to the MenT toxins following antitoxin binding could indeed position respective phosphoacceptors in range of bound substrates for phosphorylation to occur (Figs. 6 and 7). We have not been able to find evidence in the literature for auto-phosphorylation in other NTases, so the proposed catalytic mechanism of antitoxicity for MenAT1 and MenAT3 implies an additional form of NTase regulation. We hypothesize that due to the observed variability in regulatory binding partners, and the ubiquity of COG5340-DUF1814 TA modules in bacterial and archaeal genomes[36], this could be a much more widespread mechanism worthy of further exploration. For example, DUF2253 and DUF4849 proteins are both members of the DUF1814 superfamily that have been proposed to function as putative NTases[37], both of which are typically encoded immediately upstream of predicted transcriptional regulators[8]. Analysis of the Conserved Domain Database[38] revealed strict conservation of a serine or threonine at the site of MenT phosphorylation in all retrieved hits for either COG protein family. One such hit, PygT, encodes a DUF2253 domain and belongs to the functional PygAT TA system from *Pyrococcus yayanosii*. Whilst the toxin has been shown to be activated in response to high hydrostatic pressure[39], the mechanism by which its activity is counteracted remains unknown. Collectively, our findings may help shed light onto the exact mechanisms by which functionally active COG5340 and COG4861 antitoxins negate DUF1814 toxin activity across bacterial and archaeal genomes alike.

We had previously concluded that the formation of a toxin-antitoxin complex, and the resultant sequestration of MenT1, explained the mechanism of antitoxicity for MenAT1 akin to an abundance of other *M. tuberculosis* type II TA systems[17,29,40]. The observed activity of MenT1 T39A, coupled with the ability of MenA1 to neutralize in vivo toxicity and inhibit in vitro NTase activity, support such a conclusion, demonstrating that T39 is not essential to antitoxicity, unlike the equivalent S78 residue of MenT3[21]. However, the requirement for higher concentrations of antitoxin to inhibit modification of tRNA by MenT1 T39A in comparison to MenT1 WT indicates phosphorylation plays a major role in MenA1 antitoxicity, as evidenced by the existence of MenT1-p in *M. tuberculosis*[27] (Fig. 5d). Control of both systems, therefore, involves two processes. Within MenAT3, the MenA3 antitoxin is unable to prevent MenT3 S78A toxicity and so auto-phosphorylation seems the only post-translational mode of regulation, alongside previous evidence for transcriptional regulation[21]. For MenAT1 there appears to be sole focus on post-transcriptional regulation as we observed both sequestration and auto-phosphorylation (Fig. 5d), and a lack of transcriptional autoregulation (Fig. 1d). Besides functioning as a pre-requisite to phosphorylation, the biological relevance of MenAT1 complex formation in a host remains unclear. There is growing evidence supporting activation of TA systems under harsh growth conditions, with toxin activation serving as a means to arrest growth and allow bacteria to evade stressors[11,14,18,41]. Reduced cellular ATP levels have been linked to states of bacterial dormancy and persistence[42], and *M. tuberculosis* ATP concentrations have been shown to be reduced by fivefold under hypoxic conditions, with the bacterium exhibiting reduced metabolic activity[43]. Under such conditions, the requirement for activated toxin to attenuate bacterial growth would be favored by depleted pools of cellular ATP, with the inability of ADP to generate significant levels of phosphorylated toxin supporting the hypothesis that a surplus of NTPs are required for inactivation of the toxin in vivo. Clearly, there are biological requirements for different methods of toxin control, depending on how toxins are used in cells and whether they have input to the regular housekeeping of central metabolism, as we have recently described[15].

Like MenT1 and MenT3, MenT4 is a putative NTase with broad target specificity that preferentially binds GTP to modify tRNAs[20]. We have been unable to identify an antitoxic modality for MenT4 neutralization, with no detectable complex formation during SEC (Fig. 1b), an absence of phosphorylation when co-expressed with MenA4 (Fig. 1c), and no reported phosphopeptides in vivo[27]. This suggests that MenAT4 is again regulated differently from MenAT1 and MenAT3.

The mechanism by which MenT toxins are phosphorylated has not previously been understood, but here we propose a general mechanism based on antitoxin-induced auto-phosphorylation. This accounts for different families of antitoxin, neither of which show classic kinase folds, functioning to induce phosphorylation and inhibit toxins. Stable complex formation between toxin and antitoxin proteins is reminiscent of extensively characterized type II systems, whilst post-translational modification is a hallmark of the recently classified type VII systems. The existence of MenAT1 that bears features of either class lends to the idea that, while all type VII systems must somehow interact to allow for enzymatic modification of the toxin to occur, the stability of these interactions are variable and differ even between related systems. As such, our framework for classification by type requires additional nuance. Collectively, this work reveals additional layers of regulation controlling translation within our most deadly bacterial pathogen, *M. tuberculosis*. The results are expected to have wider impact due to the prevalence of the DUF1814 family and suggest an alternative mode for the regulation of ubiquitous and essential NTases.

## Methods
### Bacterial strains
*E. coli* strains BL21 (λDE3) (Novagen), BL21 (λDE3) *ΔslyD*[6], and DH5α (Invitrogen) were routinely grown at 37 °C in 2x YT media supplemented with, when necessary, 100 μg ml⁻¹ ampicillin (Ap), 50 μg ml⁻¹ spectinomycin (Sp), isopropyl-β-D-thiogalactopyranoside (IPTG, 1 mM), L-arabinose (L-ara, 0.1% w/v) or D-glucose (D-glu, 0.2% w/v). *M. smegmatis* mc²-155 (ATCC 700084) was routinely grown at 37 °C or 30 °C in LB supplemented with, when necessary, 50 μg ml⁻¹ streptomycin (Sm), 50 μg ml⁻¹ kanamycin (Km), 0.05% v/v Tween-80, 0.2% v/v glycerol, or 100 ng ml⁻¹ anhydrotetracycline (ATc).

### Plasmid constructs
Plasmids are described in Supplementary Table 1. All cloning was performed by GenScript Biotech Ltd unless stated otherwise.

### Bacterial growth assays
In vivo toxicity assays in *M. smegmatis* were performed as follows. Cultures of mc²-155 strain were co-transformed with either pGMC-vector, -MenT1, -MenT1 T39A, or -MenT1 T39C, and either pLAM12-vector or -MenA1. Serial dilutions of transformant cultures were grown on LB agar plates supplemented with 50 μg ml⁻¹ kanamycin (Km) and 50 μg ml⁻¹ spectinomycin (Sp), either in the presence or absence of 100 ng ml⁻¹ anhydrotetracycline (ATc) and 0.2% acetamide (Ace) for toxin and antitoxin expression, respectively. Plates were incubated for 3 days at 37 °C before imaging.

In vivo toxicity assays in *E. coli* were performed as follows. *MenA3* and *MenT3* were previously cloned into pTA100 and pBAD30 as described[6]. Antitoxin mutant constructs were generated by GenScript Biotech Ltd as described in Supplementary Table 1. *E. coli* DH5α were co-transformed with pBAD30 empty vector or pPF657 (MenT3 WT), and either pTA100 empty vector, pPF656 (MenA3 WT), or pTRB669-674 and pTRB717-720 (MenA3 mutants). Single transformants were used to inoculate 5 ml LB supplemented with Ap, Sp and D-glu and grown overnight at 37 °C with 180 rpm shaking. The following morning, cultures were re-seeded into fresh LB and grown to mid-log phase (OD₆₀₀ - 0.6). Samples were then normalized to an OD₆₀₀ of 1.0 by resuspending varying amounts of culture in PBS, then serially diluted 10⁻²-10⁻⁹ and spotted on M9A plates containing Ap and Sp, and either

D-glu, L-ara, IPTG, or L-ara and IPTG for repression of toxin, induction of toxin, induction of antitoxin, or induction of toxin and antitoxin expression, respectively. Plates were incubated at 37 °C for 48 h, after which they were imaged and colonies counted to determine CFU/ml.

### Protein expression and purification

Expression and purification of MenT$_3$ and MenT$_4$ was performed as previously described[6]. For large-scale expression of MenA$_1$ and MenT$_1$ for biochemistry and crystallography, *E. coli* BL21 (λDE3) Δ*slyD* was transformed with pTRB617 and pTRB629, respectively. Mutant derivatives were expressed by transforming *E. coli* BL21 (λDE3) Δ*slyD* with plasmids pTRB655 (MenA$_1$ N1-32), pTRB704 (MenA$_1$ L14R/V19R), pTRB698 (MenT$_1$ T39A), pTRB699 (MenT$_1$ D41A), pTRB700 (MenT$_1$ K137A), or pTRB701 (MenT$_1$ D152A). For toxin-antitoxin co-expression, *E. coli* ER2566 was transformed with either pTRB629 and pTRB597 (MenA$_1$-MenT$_1$), pTRB517, pPF656, and pRARE (MenA$_3$-MenT$_3$), or pTRB544 and pPF658 (MenA$_4$-MenT$_4$).

Single colonies were used to inoculate 130 ml 2x YT for overnight growth at 37 °C with 180 rpm shaking. Starter cultures were re-seeded 1:100 v/v into 2 L baffled flasks containing 1 L 2x YT supplemented with the relevant antibiotic(s) and were subsequently incubated at 37 °C until reaching an OD$_{600}$ of 0.3. At this point, incubation temperature was reduced to 21.5 °C until expression cultures reached an OD$_{600}$ of 0.55. Flasks were then supplemented with the relevant inducing agent(s) and incubated overnight at 18 °C. Cells were harvested by centrifugation at 4200 × *g* for 15 min at 4 °C, then serially resuspended in ice-cold A500 buffer (20 mM Tris HCl pH 7.9, 500 mM NaCl, 30 mM imidazole, 10% glycerol). Resuspended cells were disrupted by sonication (45% amplitude, 10 s pulse intervals, 2 min) and clarified by centrifugation at 45,000 × *g* for 40 min at 4 °C. Clarified cell lysate was transferred to a chilled glass beaker on ice and applied to a 5 ml HisTrap HP column (Cytiva) pre-equilibrated in A500. The HisTrap column was then washed with 50 ml A500, followed by 50 ml A100 buffer (20 mM Tris HCl pH 7.9, 100 mM NaCl, 10 mM imidazole, 10% glycerol), with bound proteins eluted directly onto a pre-equilibrated 5 ml HiTrap Q HP column using B100 (20 mM Tris HCl pH 7.9, 500 mM NaCl, 250 mM imidazole, 10% glycerol). The Q HP column was re-equilibrated with 50 ml A100 and transferred to an Åkta™ Pure (Cytiva), with target protein eluted by anion exchange chromatography (AEC) using a salt gradient from 100% A100 to 60% C1000 (20 mM Tris HCl pH 7.9, 1 M NaCl, 10% glycerol). Chromatographic peak fractions were analysed by SDS-PAGE to confirm the presence of target protein, then pooled and incubated overnight at 4 °C in the presence of human sentrin/SUMO-specific protease 2 (hSENP2) to facilitate cleavage of the His$_6$-SUMO tag. The following day, the SENP-treated sample was applied to a second HisTrap HP column pre-equilibrated in low-imidazole A500 (10 mM imidazole), with flow-through containing untagged target protein collected on ice and subsequently concentrated by centrifugation using the appropriate MWCO Vivaspin concentrator (Sartorius). Concentrated protein samples were then applied to a HiPrep™ 16/60 Sephacryl® S-200 HR column (S-200; Cytiva) pre-equilibrated in sizing buffer (50 mM Tris HCl pH 7.9, 500 mM KCl, 10% glycerol) and further purified SEC, with the exception of MenA$_1$, which was sufficiently pure following the second HisTrap purification step. As with AEC, SEC peak fractions were analysed by SDS-PAGE, then concentrated as before and quantified using a NanoDrop 2000 Spectrophotometer (Thermo Fisher). Final purified samples were either resuspended in A500 for immediate use, or a 1:2 mixture of storage:sizing buffer (Storage buffer; 50 mM Tris HCl pH 7.9, 500 mM KCl, 70% glycerol) for storage at −80 °C.

To purify and isolate MenT$_1$-p, both MenA$_1$ and MenT$_1$ were purified in parallel until immediately after the AEC step, then separately concentrated and resuspended in A500. MenT$_1$ was treated with hSENP2 overnight to facilitate removal of the hexahistidine tag. Equal volumes of each sample were then directly mixed at a 2:1 mole ratio of T:A in the presence of 1 mM CTP and 10 mM MgCl$_2$, with the resultant mixture incubated at 4 °C overnight. The next day, this sample was applied to a second HisTrap column pre-equilibrated in A500, with flow-through containing untagged MenT$_1$-p collected on ice. The column was then washed with 50 ml A500 to collect residual MenT$_1$-p followed by 50 ml B500 to elute His$_6$-SUMO-MenA$_1$; this sample was subsequently discarded. MenT$_1$-p was then concentrated and either used immediately or stored as described above.

To purify the MenA$_1$:MenT$_1$ complex, respective proteins were purified separately in parallel until immediately prior to the final SEC purification step. 500 µl containing 200 nmol MenT$_1$ and 500 µl containing 100 nmol MenA$_1$ were co-incubated at 4 °C overnight, then concentrated tenfold to a volume approximately equal to 0.5% geometric column volume and directly applied to the appropriate SEC column for analysis.

### β-galactosidase activity assays

Protocols for the preparation of electrocompetent *M. smegmatis* and electroporation have been previously described[44]. SAPPHIRE 2 was first used to identify regions upstream of the *rv0078B-rv0078A* transcriptional start site that may serve as promoter elements[45]. The *rv1960c-rv1959c* promoter was selected as a positive control based on previous assays demonstrating negative transcriptional autoregulation of this promoter by the ParDE1 complex[29]. 1000 bp upstream promoter regions of *rv0078B-rv0078A* and *rv1960c-rv1959c* were cloned as BamHI/KpnI digested inserts into the *LacZ* fusion plasmid pJEM15 cut with the same enzymes. Electrocompetent *M. smegmatis* mc²155 were co-transformed with either pJEM15 -vector, -P$_{rv0078B/A}$, or -P$_{rv1960c/1959c}$, and either pGMC -vector, -MenT$_1$, -MenA$_1$, -MenAT$_1$, or -ParDE1. Transformants were subjected to blue/white screening for β-galactosidase activity by plating onto LB agar supplemented with Km (50 µg ml⁻¹), Sp (100 µg ml⁻¹), Tween-80 (0.05% v/v), IPTG (1 mM), and 5-bromo-4-chloro-3-indolyl β-D-galactopyranoside (X-Gal, 40 µg ml⁻¹) in the presence or absence of ATc (100 ng ml⁻¹). Single colonies were used to inoculate 5 ml LB media supplemented with Tween-80 (0.05% v/v), glycerol (0.2% v/v), and the relevant selection antibiotics in the absence and presence of ATc (100 ng ml⁻¹). Cultures were grown at 37 °C until saturation, re-seeded 1:50 v/v in fresh media, and grown overnight at 30 °C until reaching an OD$_{600}$ of 0.8. Measurement of β-galactosidase activity was performed as described[46], with several amendments to the protocol. Briefly, 2 ml of each culture was incubated on ice for 20 min to arrest growth, then centrifuged at 4000 × *g* for 10 min at 4 °C. Cells were resuspended in 2 ml of chilled Z buffer (6 mM NaH$_2$PO$_4$.H$_2$O, 10 mM KCl, 50 mM BME, 1 mM MgSO$_4$, pH 7.0), with 1.5 ml aliquoted into a clean cuvette for measurement of OD$_{600}$. The remaining cells were then diluted 1:2 in 0.5 ml fresh Z buffer and lysed by mechanical homogenization using a FastPrep-24™ 5G bead beating grinder and lysis system (MP Biomedical). Cells were briefly vortexed and incubated at 28 °C for 5 min. Reactions were initiated following the addition of 100 µl ONPG (4 mg ml⁻¹) and were allowed to proceed for 15 min at 30 °C before terminating with 200 µl Na$_2$CO$_3$ (1 M). Reactions were then centrifuged at 16,000 × *g* for 5 min to remove cellular debris, with 500 µl of supernatant transferred into a clean cuvette and diluted with 500 µl Z-buffer for measurement of OD$_{420}$ and OD$_{550}$ values and subsequent calculation of activity as described[46].

### Protein crystallization and structure determination

Purified MenT$_1$-p and native MenT$_3$ protein samples were concentrated to 12 mg ml⁻¹ in Crystal buffer (20 mM Tris HCl pH 7.9, 150 mM NaCl, 2.5 mM DTT) and crystallization screens were performed using a Mosquito Xtal3 robot (SPT Labtech) using the sitting drop method, with 200:100 nl and 100:100 nl protein:condition drops set for each condition screen. MenT$_1$-p formed thick cuboid crystals in condition E9 (4 M Sodium Formate, 0.1 M Tris, pH 7.5) and thin rod-shaped needles in condition G9 (4 M Sodium Formate, 0.1 M Tris, pH 8.5) of Clear Strategy II Eco HT-96 (Molecular Dimensions). MenT$_3$ formed

thick, six-sided needles in condition G5 (0.2 M calcium acetate hydrate, 0.1 M Tris pH 8.5 and 25% w/v PEG 2000 MME) of the same screen. To harvest crystals for subsequent structural determination, 20 µl screen condition was mixed with 20 µl Cryo Buffer (25 mM Tris HCl pH 7.9, 187.5 mM NaCl, 3.125 mM DTT, 80% glycerol), then added to the protein crystal drop at a 1:1 v/v ratio. Crystals were then immediately extracted from the drop using the appropriately sized nylon loop and transferred to a unipuck immersed in liquid $N_2$.

Diffraction data were collected at Diamond Light Source on beamlines I04 ($MenT_1$-p) and I24 ($MenT_3$) (Table 1). Two 360° datasets were collected for $MenT_1$-p at 0.9795 Å and merged using iSpyB (Diamond Light Source). A single 720° dataset was collected for $MenT_3$ at the same wavelength. Data were processed using AIMLESS from CCP4[47,48] to corroborate spacegroups. Both structures were solved by PHASER MR[49], with $MenT_1\alpha$ from the $MenT_1$ crystal structure (PDB 8AN4) and $MenT_3$-p (PDB 6Y5U) used as search models for $MenT_1$-p and $MenT_3$, respectively. The structures were further built using REFMAC in CCP4[50], then iteratively refined and built using COOT[51] (Ramachandran statistics; $MenT_1$-p 91.62% favored, 8.38% allowed, 0.00% outliers; $MenT_3$ 97.90% favored, 2.10% allowed, 0.00% outliers). The quality of the final models was assessed using COOT and the wwPDB validation server[52]. Structural figures, including alignments and superpositions, were generated using PyMol[53].

## Cell-free protein synthesis

A cell-free transcription/translation coupled assay (PURE system, Protein synthesis Using Recombinant Elements, NEB) was used to monitor the effects of $MenT_1$ and $MenT_1$-p on protein synthesis as previously described[6,20]. Template DNA of *gfp* was added to the PURE system according to the manufacturer's instructions, either in the absence or presence of the toxin. To pre-incubate $MenT_1$ with tRNA, a PURExpress (Δaa, tRNA) kit (NEB, E6840S) was used. 0.33 µl tRNA from the kit was incubated with 0, 2.5, 5, and 10 µM $MenT_1$ or $MenT_1$-p for 3 h at 37 °C, then the assay was performed as per the manual, except 1.5 µl pre-incubated tRNA was used in a 5 µl reaction volume. Following pre-incubation, protein synthesis was performed for 2 h at 37 °C, prior to separation of samples on 4–20% miniprotean TGX gels by SDS-PAGE (Bio-Rad). Gels were subsequently analysed by western blots by probing with monoclonal anti-GFP antibody (ThermoFisher MA5-15256; dilution 1/3000), detected using HRP-conjugate anti-mouse IgG (H+L) secondary antibody (Promega W4021; dilution 1/2500) and visualized by Image Lab software (Bio-Rad).

## In vitro transcription of tRNAs with homogeneous 3′ ends

An optimized version of the hepatitis delta virus (HDV) ribozyme was used to generate homogeneous tRNA 3′ ends as described. Briefly, the DNA template T7-tRNA-HDV was amplified from plasmid pUC-57Kan-T7-tRNA-HDV (Supplementary Table 1). Labelled tRNAs were prepared by in vitro transcription of PCR templates using T7 RNA polymerase. The T7 RNA polymerase transcription reactions were performed in 25 µl total volume, with a 5 µl nucleotide mix of 2.5 mM ATP, 2.5 mM UTP, 2.5 mM GTP, 60 µM CTP (Promega, 10 mM stock) and 2–4 µl 10 mCi ml$^{-1}$ of radiolabelled CTP [α-$^{32}$P]. 50 to 100 ng of template were used per reaction with 1.5 µl rRNasin 40 µ ml$^{-1}$ (Promega), 5 µl 5× optimized transcription buffer (Promega), 2 µl T7 RNA polymerase (20 µ ml$^{-1}$) and 2.5 µl 100 mM DTT. Unincorporated nucleotides were removed by Micro Bio-Spin 6 columns (Bio-Rad) according to manufacturer's instructions. The transcripts were gel-purified on a denaturing 6% acrylamide gel and eluted in 0.3 M sodium acetate overnight at 20 °C. The supernatant was removed, ethanol precipitated and resuspended in 14 µl nuclease-free water. Radioactively labelled tRNAs carrying a 2′,3′ cyclic phosphate at the 3′ end was dephosphorylated using T4 polynucleotide kinase (NEB) in 100 mM Tris-HCl pH 6.5, 100 mM Mg(CH$_3$COO)$_2$ and 5 mM BME in a final volume of 20 µl for 6 h at 37 °C. All assays were desalted by Micro Bio-Spin 6 columns (Bio-Rad).

## In vitro tRNA modification assays

$MenT_1$ NTase activity was assayed in 10 µl reaction volumes containing 20 mM Tris-HCl pH 8.0, 10 mM $MgCl_2$, and 1 µCi µl$^{-1}$ of radiolabelled rCTP [α-$^{32}$P] (Hartmann Analytic) and incubated for 2 h at 37 °C. 1 µg total RNA from *M. smegmatis* was used per assay with 5 µM of protein. Reactions were purified with Bio-Spin 6 Columns (Bio-Rad) and mixed with 10 µl of 2× RNA loading dye (95% formamide, 1 mM EDTA, 0.025% SDS, xylene cyanol and bromophenol blue), denatured at 90 °C and separated on 6% polyacrylamide-urea gels. The gel was vacuum dried at 80 °C, exposed to a phosphorimager screen and revealed by autoradiography using a Typhoon phosphorimager (GE Healthcare).

## In vitro inhibition of toxin activity

$MenA_1$ antitoxin activity was assayed using either in vitro-transcribed tRNA Met-2 or 1 µg total RNA from *M. smegmatis* as substrates. For the co-incubation assay, $MenT_1$ or its derivatives (5 µM) and increasing molar ratios of $MenA_1$ were incubated with target substrate and 1 mM CTP in 10 µl reaction volumes supplemented with 20 mM Tris-HCl pH 8.0 and 10 mM $MgCl_2$. All reactions were incubated for 4 h at 37 °C.

## Thermal shift assays (TSA)

Thermal shift assays were performed to assess the ability of target proteins to bind nucleotide substrates, with protein-ligand interactions evidenced by changes in $T_m$[54]. $MenT_1$ and its derivatives or $MenA_1$ were first labelled with $4 \times 10^{-3}$ µl SYPRO orange dye (ThermoFisher) per 1 µl protein. Reactions comprising varying concentrations of protein(s), 1 mM NTP, and 10 mM $MgCl_2$ were supplemented with 10 µl TSA buffer (20 mM $NaH_2PO_4$, 100 mM NaCl, pH 7.4) and made up to 20 µl with nuclease-free water. Samples were co-incubated overnight at room temperature in sealed 96-well semi-skirted PCR plates (Starlab), then centrifuged briefly to collect liquid and inserted into a CFX connect real-time qPCR machine for thermal shift analysis. Protein denaturation was performed by incrementally increasing temperature from 25 to 95 °C. Deconvolution of thermal shift isotherms was performed using the NAMI python tool[55]. Melt curves and thermal shift graphs were generated using Prism (GraphPad).

## Mass spectrometry

Purified protein samples were buffer exchanged into 10 mM ammonium bicarbonate using the appropriate MWCO spin concentrator and submitted for positive ion electrospray time-of-flight mass spectrometry (ES$^+$-ToF MS) at a final concentration of 0.5 mg ml$^{-1}$. Samples were desalted online using a MassPrep On-line Desalting Cartridge (Waters, UK) prior to mass spectrometry. Chromatography was performed at a flow rate of 0.4 ml min$^{-1}$ utilizing a 0.1% v/v formic acid:acetonitrile gradient. Measurements were obtained using a Xevo QToF (Waters, UK) mass spectrometer ran in full scan mode scanning between 500 and 2000 u in 1 s. The electrospray capillary, sampling cone and extraction cone were at 3 kV, 30 V and 5 V, respectively. Source temperature and desolvation temperature were held at 120 °C and 500 °C and the cone gas and desolvation gas flow rates were 20 L hr$^{-1}$ and 800 L hr$^{-1}$, respectively. The instrument was calibrated externally using sodium formate and individual measurements corrected using a leucine enkephalin 'lockmass' solution delivered using a second electrospray probe. Data was deconvoluted to give neutral masses using MassLynx 4.1 and MaxEnt 1 (Waters, UK).

To conduct LC-MS/MS analysis, $MenT_1$ samples were prepared as described above and submitted to Durham University's in-house Proteomics facility for ProAlanase digests and subsequent analyses.

## Phos-Tag SDS-PAGE

Incubation mixtures comprising combinations of either 10 µM $MenT_1$, 5 µM $MenA_1$, 1 mM NTPs, or 10 mM $MgCl_2$ were made up to

10 μl with nuclease-free water. Samples were incubated overnight (temperature varied between experiments), then directly mixed with 10 μl 2× Phos-Tag loading dye (125 mM Tris-HCl pH 6.8, 10% v/v BME, 0.002% BPB, 4% w/v SDS, 20% glycerol) and boiled at 95 °C for 5 min. Samples were cooled to room temperature and loaded onto Phos-Tag SuperSep 15% pre-cast acrylamide gels (Wako pure industries) immersed in 1× Tris Glycine running buffer. Electrophoresis was performed at 180 V constant at 4 °C until the dye front reached the end of the gel, at which point gels were removed and stained with Coomassie for 2 h. Gels were de-stained in water overnight prior to densitometric analysis of band intensity using ImageJ software[56], with background normalization and subtraction using the rolling-ball method (radius set to 50 pixels).

### Phosphorylation assays
Incubation mixtures were prepared as described for Phos-Tag SDS-PAGE, with the exception of making all samples up to 100 μl final volume. Following overnight incubation at RT, samples were buffer exchanged into 10 mM ammonium bicarbonate and submitted for mass spectrometry analysis as described above.

### Analytical size exclusion chromatography
A Superdex™ 75 increase 10/300 GL SEC column (S-75i; Cytiva) connected to an Åkta™ Pure FPLC system (Cytiva) was pre-equilibrated in two column volumes of analytical SEC buffer (20 mM Tris-HCl pH 7.9, 150 mM NaCl). Protein samples were made up to 120 μl final volume with nuclease-free water and comprised varying amounts of protein. A 100 μl capillary loop was first washed with 500 μl nuclease-free water followed by 500 μl analytical SEC buffer before and between each run using a 500 μl Hamilton syringe, with samples loaded onto the pre-equilibrated loop using a 100 μl Hamilton syringe. Samples were applied by running 1.2 column volumes of analytical SEC buffer through the capillary loop at flow rate of 0.5 ml min$^{-1}$; the excess buffer serving to pre-equilibrate the column for the following run. In instances where the content of chromatogram peaks required verification by SDS-PAGE, 0.5 ml fractionation was performed, and peak fractions were collected in 96-well deep-plate blocks.

Calibration curves were generated by plotting the elution volumes ($V_e$) of controls from LMW/HMW calibration kits (GE healthcare) against their respective known molecular weights ($M_r$). Calibration samples were prepared in 2 individual mixtures, Mix A (3 mg ml$^{-1}$ RNase A, Conalbumin, Carbonic Anhydrase, Aprotinin) and Mix B (3 mg ml$^{-1}$ RNase A, Aprotinin, 4 mg ml$^{-1}$ Ovalbumin), made up to a final volume approximately equal to 0.5% geometric column volume. For determination of column void volume ($V_o$), 1 mg ml$^{-1}$ Blue Dextran was applied to the column as above, with elution volume directly proportional to $V_o$. Elution volumes ($V_e$) were calculated using the Peaks function in Unicorn™ 7 (Cytiva) and converted to partitioning coefficients ($K_{av}$) using the following equation:

$$K_{av} = \frac{V_e - V_o}{V_c - V_o} \qquad (1)$$

Molecular weight and Stokes radius calibration curves were subsequently plotted using Prism (GraphPad) as $K_{av}$ vs Log$_{10}$($M_r$, kDa) and Log$_{10}$($R_{st}$, Å) vs $K_{av}$, respectively. Observed $R_{st}$ values were generated by performing linear regression on respective plots using the following equations:

$$M_r = 10^{\wedge}\left(\frac{K_{av} - c}{m}\right) \qquad (2)$$

$$R_{st} = 10^{\wedge}((m(K_{av}) + C) \qquad (3)$$

Observed values were then compared against calculated hydrodynamic radii. Radius calculations of inputted crystal structures or AlphaFold predictive models[57] were performed using the HullRad tool (Fluidic Analytics)[58].

### Molecular docking
To generate predicted poses for each NTP, the MenT$_1$ structure was retrieved from the Protein Data Bank (PDB 8AN4) and uploaded directly to PyMol and UCSF Chimera for visual inspection and preparation such as deleting the solvent and non-complexed ions. Prepared structures were uploaded to SeeSAR 13.0 (BioSolveIT)[59], and "druggable" binding sites for CTPs were mapped using SeeSAR-Pocket. The sites with the highest consensus scores (DoGSiteScore > 0.49)[60], largest accessible volumes (>750 Å$^2$) and number of H-bond donors and acceptors mapped (>20 and >25 for each site, respectively), were selected as targets for molecular docking calculations.

Molecular docking calculations were performed using HYDE scoring function, which combines intrinsic interactions and desolvation energies in protein-ligand complexes[61]. Calculations were carried out using the unrestricted non-covalent mode with 500 poses per site, with high clash tolerance and no ring puckering enabled. The five best-scored poses for each ligand, including those with best calculated binding affinity, low torsional strain and lack of intramolecular clashes, were selected for subsequent analysis and the consensus pose (Supplementary Fig. 9H) went into molecular dynamics simulations.

The binding sites and molecular docking poses selected were subsequently validated by the recently published crystal structure of MenT$_3$ bound with CTP (PDB 8XHR). The predicted NTP binding site and poses for MenT$_1$ overlap with the CTP binding site and mode (RMSD$_{NTP}$ = 1.37 Å, Supplementary Fig. 9H, I). Molecular docking of CTP to MenT$_3$ (positive control) reproduced the experimental binding mode very well (RMSD$_{CTP}$ = 1.14 Å; Supplementary Fig. 9J).

### MD simulations
All-atom MD simulations used GROMACS[62] with AMBER99SB-ILDN[63–65] parameters. Each simulated complex was immersed in a cubic TIP3P water box, set to be 1 nm away from the edge of the protein[66]. 3D-PBC were applied. Na$^+$ and Cl$^-$ ions were added to maintain the charge neutrality. Each system underwent energy minimization via the steepest descent method for 1000 cycles and the conjugate gradient method for further refinement, with the energy step size of 0.001 nm and a maximum of 50,000 steps. The minimization was concluded when the maximal force descended below 1000 kJ/mol/nm. Long-range electrostatic interactions were addressed using the Particle-Mesh Ewald (PME) method[67], while a cut-off of 1.0 nm was applied for short-range electrostatic and van der Waals interactions. Following energy minimization, all systems were subjected to a 500 ps NVT equilibration with a step size of 2 fs. The protein and non-protein groups were gradually heated to 300 K under the influence of the V-rescale thermostat[68] with a time constant of 0.1 ps. LINCS (Linear Constraint Solver) position restraints[69] were applied to the bond lengths and angles of the backbone atoms. The non-bonded short-range interactions were treated with the Verlet cut-off scheme, setting the cut-off distance to 1.0 nm. Long-range electrostatics were once again addressed with PME. Subsequently, NPT equilibration was performed, where temperature was maintained at 300 K with the continued utilization of the temperature coupler, followed by the initiation of Parrinello–Rahman pressure coupling[70] for 500 ps of pressure equilibration, with the target pressure established at 1 bar. Post-equilibration, the systems were subjected to three separate 300 ns simulations to obtain trajectories for analysis, discarding the initial 10 ns of data during the final evaluation. Trajectory analyses were conducted using GROMACS tools. The overall stability was evaluated by root mean square deviation (RMSD), and local flexibility was assessed via per-residue root mean square fluctuation (RMSF).

Principal component analysis (PCA) was employed to explore the key motion modes. The protein structural stability in terms of folding was analysed through the radius of gyration (Rg). Hydrogen bond examination was carried out utilizing VMD[71]. Protein-protein interaction enthalpy calculations were performed using parameters derived from AMBER parm99 classical molecular mechanical force fields and a GB/SA implicit solvation model. All calculations were performed using INTAA server[72].

To explore the interaction energies in phosphorylated and non-phosphorylated models, we utilized Plumed 2.90[73] to implement well-tempered metadynamics simulations. Four distinct collective variables were defined to measure distances between centroids for different protein-protein interfaces. The initial height of each Gaussian potential (HEIGHT) was set to 0.6 kJ/mol, with a width (SIGMA) of 0.1 nm, and a Gaussian added every 200 simulation steps (PACE). A bias factor (BIASFACTOR) of 10 was established to realize a well-tempered sampling strategy. The addition of Gaussians was based on the current value of the collective variable and was recorded in the HILLS file, to facilitate subsequent analysis and reproduction of the simulation process. Free energy profiles were obtained by integrating data collected during the metadynamics simulations. These profiles displayed the change in free energy as MenT$_1\alpha$ and MenT$_1\beta$ dissociate from MenA$_1$ in both phosphorylated and non-phosphorylated states. Comparing the free energy minima and peaks of each allows comparison of interaction stabilities between MenT$_1\alpha$ and MenT$_1\beta$ with MenA$_1$ under different phosphorylation states.

### Reporting summary

Further information on research design is available in the Nature Portfolio Reporting Summary linked to this article.

## Data availability

The crystal structures have been deposited in the Protein Data Bank under accession numbers 8RR5 (MenT$_1$-p) and 8RR6 (MenT$_3$). Other PDB entries used in this study can be found under the following accession numbers: 8AN4 (MenT$_1$); 8AN5 (MenAT$_1$ complex); 6Y5U (MenT$_3$-p); 8XHR (MenT$_3$-CTP); 6Y56 (MenT$_4$); 6Y8Q (AbiEi). All other data needed to evaluate the conclusions in the paper are present in the paper and/or Supplementary Information/Source Data file. Source data are provided with this paper.

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

## Acknowledgements

We gratefully acknowledge Diamond Light Source for time on beamlines I04 and I24 under proposal MX24948. We thank Dr. Adrian Brown for

performing peptide digests and subsequent LC-MS/MS analyses, and Jenny Readshaw for assisting with preparation of *M. smegmatis* liquid cultures. We thank Dr. Matthew Kitching and Dr. Michael Carroll for reviewing the proposed phosphorylation mechanism. This work was supported by the Engineering and Physical Sciences Research Council Molecular Sciences for Medicine Centre for Doctoral Training [grant number EP/S022791/1] to T.J.A and A.K.B; the Programme d'Investissements d'Avenir [grant number ANR-20-PAMR-0005] to P.G.; the Fondation pour la Recherche Médicale grant [EQU202403018015] to P.G.; the National Natural Science Foundation of China, [grant number 32000021] to X.X.; and a Springboard Award from the Academy of Medical Sciences [grant number SBF002\1104] to B.U. and T.R.B. For the purpose of open access, the author has applied a Creative Commons Attribution (CC BY) licence to any Author Accepted Manuscript version arising from this submission.

## Author contributions

Analysed data: T.J.A., X.X., S.X., B.U., P.S., M.G., A.K.B., P.G. and T.R.B. Designed research: T.J.A., X.X., S.X., B.U., A.K.B., P.G. and T.R.B. Performed research: T.J.A., X.X., S.X., B.U., M.G., A.K.B. Wrote the paper: T.J.A. and T.R.B. with contributions from all the authors. Funding acquisition: A.K.B., P.G., and T.R.B. Supervised the study: A.K.B., P.G., and T.R.B.

## Competing interests

The authors declare no competing interests.
