## [Peer Review File · Nature Communications]

Reviewers' Comments:

Reviewer #1:

Remarks to the Author:

Toxin/antitoxin (TA) systems are ubiquitous and diverse regulatory elements that bacteria use to correspond to different environmental stress conditions and in plasmid and bacteriophage defense. The toxins are typically blocking transcriptional and translational processes resulting in reduced energy consumption, cell division and may induce dormancy and allow, for example, bacteria to escape antibiotic treatment. The partner antitoxin typically prevents unwanted toxin activity under 'normal' conditions, typically by forming a stable inactive complex (sequestration) or modifying the toxin. This particular work deals with the nucleotidyltransferase (NTase) family of toxins that block bacterial translation by tRNA processing. *M. tuberculosis* carries 4 gene pairs coding a MenAT system, wherein the toxin MenT is a NTase. Recent work of the author's group provided structural insights in the activity of the toxin NTase, but how the antitoxin MenA counteracts the toxin activity was not yet understood. Curiously, while the 4 MenT genes are largely similar, the antitoxins are more diverse.

In this manuscript, Arrowsmith present an entirely new mechanism by which MenA1 and MenA3 induce autophosphorylation of the toxin. They present convincing biochemical, structural biological and biophysical experiments together with molecular dynamics approaches to support the insight that binding of the antitoxin provokes subtle movements in the MenT1 and MenT3 active site to present a threonine or serine to the gamma phosphoryl of a bound nucleotide and mediate a phosphotransfer to the amino acid. This phosphorylation provokes an electrostatic repulsion and prevents binding of a substrate nucleotide for the NTase action.

Inhibitory autophosphorylation was previously also described as a mechanism for toxin inhibition, for example for the HipA toxin in *E. coli*. However, in this case, phosphorylation occurs in the absence of the antitoxin and is therefore mechanistically different. As such, this manuscript presents a completely novel mechanism of toxin activity regulation induced upon sequestration by the antitoxin. The authors present new insights in translational control in *M. tuberculosis*, the world most infectious killer according to WHO, that just recently published a report disclosing the highest number of people diagnosed with tuberculosis since monitoring started in 1995. Fundamental studies to unravel molecular processes in *M. tuberculosis* are thus highly demanded. Technically, the manuscript is excellent. The data provided are highly supporting the proposed mechanism and are presented nicely.

There are a few, rather minor comments to be addressed.

On line 183, a number of mutations are listed that are demonstrated to abolish the phosphorylation. However, the authors do not provide a rationale for the choice of these mutants. There is no reference to previous work on these mutations, for example in the introduction. If the mutants are based on the structural biology work provided in this

manuscript, this should be more clearly addressed.

On page 9, LC-MS data are used to demonstrate that T39 is phosphorylated. The supplementary figure indicates that this is most likely based on analysing a tryptic digest peptide mixture. This is not clearly mentioned, nor is this included in the method section.

On page 17, line 366-373, the authors discuss the movements of some residues. The authors should be more consequent in addressing whether this refers to MenT1beta for clarity.

The supplementary movie is not entirely clear, it may require some extra information. As it is presented now, it starts with a dissociation event followed by a reassociation, while the reverse is probably meant to be presented.

Lastly, in the discussion on page 22, line 518, reference is made to the relation of altered ATP levels and toxin (in)activation. However, autophosphorylation of MenT1 also occurs in the presence of ADP which raises question on the ATP concentration dependency for this particular toxin. This deserves to be discussed.

Finally, the manuscript may miss some broader context. The MenT toxins belong to a large family of NTases that is said to be widespread throughout microbial life (as mentioned in the introduction). How about MenA1 and MenA3, is the presented mechanism for MenAT1 or MenAT3 indeed as general as mentioned on line 532 ?

Reviewer #2:

Remarks to the Author:

The paper by Arrowsmith et al., addresses the mechanism of neutralisation of MenT tuberculosis toxins by sequestration and autophosphorylation. The authors used a combination of X-ray crystallography and MD, biochemistry and biophysics with classical microbiology to investigate the regulation of various types of toxins that are controlled by very different antitoxins. This is a solid and well written manuscript that deconstructs different aspects of the regulation of these toxins to extract meaningful insights into their mechanism of action and neutralisation.

The one major concern I have regarding their conclusions relates to their use of a T39A substitution and the interpretation of the results involving these results.

The authors performed several measurements with T39A including experiments aimed at qualifying the possible role of T39 in antitoxicity. The experiments of Fig 5D are used to claim that T39 is not essential for antitoxicity based on the observation that the toxin retains activity. However one can also interpret this as possible effects in the catalytic cycle of the enzyme introduced by the substitution of Thr for Ala (while tRNA is still slightly modified, the activity of the enzyme appears severely affected). To support these claims the authors should use for example a Cys substitution which is closer to Thr both in shape and properties.

Minor comments:

- 1) Analytical SEC is described in the Methods as done in a 1030 column, however the experiments from Fig. 1B are clearly from a bigger column. Could the authors please comment?
- 2) Related with this, the colours and representations from 1B make it difficult to interpret the figure especially the middle panel. In addition, the use of Absorption vs MW axes used in Fig. 5 is not a usual way of representing SEC. For example is difficult to assess from this figure the void volume of the column.
- 3) It would be helpful to show T39, D41 and K137 in Fig. 2
- 4) It would be good in Fig. 3, instead of the two views of the toxin to have a second a panel with a zoom to the TPO showing an omit map of the phospho-threonine.
- 5) While the docking experiments are an important part of the paper, they are poorly described in the Methods section.
 - a) How was the active site defined?
 - b) Can they plot the differences (in score, rmsd) between the the selected poses and the rest?
- 6) Is there a reason why the oxygen atoms of phosphate groups are not shown?
- 7) How conserved is T39 (or a possible Ser substitution) at this position?
- 8) The authors should provide the AlphaFold score of the models reported in the paper.

Reviewer #3:

Remarks to the Author:

In this work, the authors have studied nucleotidyltransferases and how in M. tuberculosis auto-phosphorylation of the nucleotidyltransferases can provide a regulatory function through modifying the toxin—anti-toxin interplay that is so crucial to the life cycle of the bacteria. In particular, they focus on nucleotidyltransferases in Mtb that are toxins which can be controlled by their corresponding antitoxins. The work is worth publishing, after the following concerns are addressed.

Major concerns.

“MenT toxins can auto-phosphorylate to control activity, catalysed by antitoxin-dependent movement of the target active site residue towards the donor phosphate.” The calculations the authors have done cannot conclusively show this. Such results should be couched as “predictions” or “calculated results”.

Also, the authors should report in more detail on their analysis of the motion of the active site residues.

The docking protocols utilized have not been explained. Instead, we are told that the structures were prepared for docking and then are told the results. That is not sufficient information for reproducibility. The choices made for docking can affect whether the results are sensible or not, so therefore it is not possible for the reader to judge the suitability of the docking methods without that information.

Fig. S9b. shows that changes are still occurring after 300 ns. Significant changes are happening even in the last 50 ns. It is better to run the simulations longer. Also, it is best not to rely on a single simulation for each set of conditions but to have at least triplicate simulations. The authors mention having run simulations in triplicate but then do not show their results. They should show the results in the supporting information, so that the readers can see how representative are the simulations presented in the main text.

Figure 1 illustrates major problems with the MS data reporting. That also affects several of the Supporting images. See the list of concerns under Fig. S1, below. For Fig. 1c:
--Axes should be labeled, a standard requirement for scientific publications. What quantity is presented on the y-axes of the charts? “percent” or “Miller units” refer to the values along the scale but: percent of what? Meanwhile, for the x-axis, “mass” is the quantity but the units should be specified or an expression of mass-to-charge ratio should be given in parentheses (M/Z).
--Notice that the signal-to-noise ratio is much better for the 2nd chart than for the first or third chart. The authors should provide an explanation about that. Also, there is a strange offset for the charts, such that the baseline is above 0. That simply looks like sloppy positioning of the axes, but may indicate a slight initial percent rise at the start of the experiment.

l.807. The equilibration is too short. Make it at least twice as long, based on standard practice of how much equilibration is needed for such a system.

Other concerns.

Fig. 1.

(d) There are two images which represent separate experiments but they are labeled identically. Instead, their overall label should be different (to match what is different between the experiments, as described in the caption).

Fig.3.

--“orange for phosphate”. Do the authors mean “orange for phosphorus”? Or is the orange stick supposed to represent the whole steric bulk of a phosphate group? In the latter case, it would be best to show the P and all its attached O’s, not just a single stick,

so that its role to provide steric bulk is more obvious from the image. As ll.224-225 state, the phosphate provides “steric occlusion of the active site”, but that does not appear clear in the current version of the figure.

L.403. Methodological choices utilized should be specified for the two computational methods mentioned in this sentence.

ll.606-607. Make sure to label properly concentration (in molar units, such as “nanomolar” or “nM”) and quantity (number of particles, such as “nanomoles”).

l.795. “poses were subsequently filtered based on ... desirable physicochemical properties”. Docking poses should not be ruled out based on desirable pchem properties. What properties were used? Why would that be a sensible approach to distinguish between poses for one particular molecule, considering that a molecule’s physicochemical properties hold true irrespective of their docking pose?

l.796. (also ll.272-273.) What is a “highest affinity” pose? Docking yields docking scores which are not the same as binding affinities. It is best to use the term “best-scored poses”.

l.796. Was the filtering done before or after the best docked pose was selected?

Fig S1.

--“+/-“ should be used instead of “-/+”

--Axes should be labeled, a standard requirement for scientific publications. What quantity is presented on the y-axes of the charts? “percent” or “Miller units” refer to the values along the scale but: percent of what? Meanwhile, for the x-axis, “mass” is the quantity but the units should be specified or an expression of mass-to-charge ratio should be given in parentheses (M/Z).

--Notice that the signal-to-noise ratio is much better for charts (b) and (c) compared to (a). The authors should provide an explanation about that. Also, there is a strange offset for (c), such that the baseline is above 0. That simply looks like sloppy positioning of the axes.

--The first lane in figure (d) looks unnecessary. Is it blank? Just 0? It appears empty. If it is important, add this to the caption. In any case, do not put the legend right above that lane, but have it centered horizontally instead, to avoid confusion.

--The lanes in (d) should be labeled more completely; note that the figure caption points out some aspects of the lane contents that are not obvious from the lane label in the chart. So for example in the last lane it is not only P + MenAT1 but also pGMC and pJEM15, right? Those extra items should be added to the lane label or else the lane label is confusing. It might be better to reduce the lane label to a simpler form (such as a

number) and then make sure the caption itself is as clear as possible.

Fig S2.

--For Fig. S2, also, it would be better to label each panel with a letter, instead of only labeling the three columns. Some of the items in column a do not correspond to what is found in columns b and c. So it can be very confusing to a reader who is trying quickly to understand what is in the figure and how things are arranged.

--Notice that the signal-to-noise ratio is much better for almost all the charts than for the third one in section (a). The authors should provide an explanation about that.

--It is not sufficient to say that each experiment was done three times and a representative image is shown. Explain more. There is lots of confusion in the literature when triplicates are mentioned but exactly what was done is not explained well. Was the MS of the same sample run three times or were all the experimental procedures repeated three times? What is the technical definition of “representative”? Quantify that.

--Under (c), three of the charts have another rather tall peak just a little higher than the main peak. Explain.

Figs S3-S5, S7. See comments on early supporting figures.

Fig S6.

--For (b), “data are representative of two independent experiments”; in what way are they representative? After that it is stated “bars display mean +/- SEM”. So was the experiment done twice but each time in triplicate? Very confusing.

--The caption has the presence of MenA1 ascribed backwards to charts (c) and (d).

Fig. S7.

--Are those best docking poses or did you impose a requirement that the LHS be superimposed? Explain more about the rank of the chosen poses for the different molecules.

--What is a “highest affinity” pose? Docking yields docking scores which are not the same as binding affinities. It is best to use the term “best-scored poses”.

--“Superposition of poses” can have two meanings. In one case, a computational procedure is used to align poses. In the other meaning, images are unchanged but are overlaid. Most likely the authors simply overlaid multiple docked structures without superposition. So the term should be an “overlay of poses”. If instead the authors did an alignment, they should justify why they did that and what relevance that is to the scientific inquiry.

Fig. S8.

--“greater than 10 less than or equal to 20” is technically incorrectly written. Could be

fixed by adding “and” in the middle (“greater than 10 and less than or equal to 20”) or could be written in a more compact form.

--Not “R_2” but “R²” (superscript).

Reviewer #4:

Remarks to the Author:

Overall, the manuscript by Arrowsmith et al. presents a convincing and highly intriguing account of how antitoxin binding induces phosphorylation in a widespread family of NTase toxins and repurposes the toxin active site from nucleotide transfer towards auto-phosphorylation in order to control activity. The study elegantly combines in vitro functional studies using a wide range of biochemical assays with structure prediction and experimental structure determination as well as molecular modelling to arrive at a cohesive and convincing molecular model. Overall, the manuscript is highly interesting and is suitable for publication following careful revision. However, in the current version, several of the figure panels are somewhat preliminary and could be improved. My main concerns therefore relate to presentation of the data as outlined below.

Major comments:

Several of the figure legends are too brief and do not provide the required information to interpret the data, e.g. 1a, 1d and S5c, see specific comments below. Please carefully check that all figure elements are described in all legends including all lanes/bars/labels.

L. 180. I am not convinced that you can conclude that Mg²⁺ is not required. Have you considered that it could be carried along by one of the protein or nucleotide components, consistent with the observation that you see increased activity with added Mg²⁺? In fact, I would be surprised if Mg²⁺ was not required.

For bar charts, I believe it is recommended to all measured data points as individual markers if n<10, not just mean +/-SEM. Since you do not have that many repetitions, I suggest you follow that standard for all bar charts.

Overall, I am disappointed with the quality of the SDS-PAGE (PhosTag) gels, most of which have run very unevenly. Sometimes, like in Fig. 2a and S6a this even makes it hard to interpret the data. What is the reason for this? We've done Phos-tag gels and I believe they are regular SDS-PAGE gels with an extra reagent added, so it should be straightforward to make them run more smoothly with commercial gels. Consider repeating some of the most important ones (like those in the main figures) for a better and more convincing overall presentation of the paper.

Figure 2f. How can the mean of a integrated band intensities be zero with a large SEM (WT + bar)? An integrated band intensity cannot be negative, so if the SEM is large (about +/- 20% here) then the mean must be positive and >0. Please check the data once more.

L. 223. For the RMSD, it appears that all atoms were used (1127)? I don't think this the best way to reveal folding changes (side chains can be floppy and are affected by crystal contacts). Instead, I would calculate RMSD using Calpha atoms only and compare the values for the entire protomer to that calculated from the loop region only to show that the loop moves upon phosphorylation.

L. 248. Isn't the difference in melting point just 1.4° between CTP and the other nucleotides? From the figures, it appears that the thermal shift is 3.8° for CTP and 2.4° for the others, so a 1.4° difference between them, please check this.

L. 272. Is there no structure of a homologous protein with nucleotide bound that you can use for comparison of the binding mode? It would be nice to confirm the docking poses. Also, I find it very odd that you observe no base-specific interactions given that the NTase prefers CTP. Can you comment on this?

Fig. 5efg. I don't believe it's appropriate to have MW on the x-axis of a SEC trace. You should always have elution volume (the true variable) and then indicate with arrows the elution volumes and MW your standard proteins. Also, it looks like Ment-p alone shifts compared to the unphosphorylated versions, can you comment on this?

Fig. S5d. Please include the unbiased Fo-Fc difference map from before modelling the phosphate group into the structure. Refined 2Fo-Fc is biased by the model and could in principle show artificial phosphate group density.

Minor comments:

Figure 1a. It is not clear from the figure if it is the N-terminal or C-terminal parts of the toxins that are conserved. I would suggest that you use dashed lines going across the isoforms or similar colours of domains to indicate conserved parts now that the figure is drawn to scale anyway. In the legend¹, what does "original and revised nomenclature" refer to?

Figure 1b. I would be more cautious about listing the estimated MW for the SEC experiments. SEC elution volumes depend on hydrodynamic radii, which are only app. corresponding to MW, so it makes no sense to list MW so precisely (5 digits). Also, please include a gel showing whether (or not) the two proteins bind and co-elute with a tag on one protein in the three cases.

Figure 1d. What does ATc +/- refer to? Also, the blue colour is very faint, so I would like to ask you to include a positive control where there is a clear repression of the lac promoter.

L. 132. It should be mentioned how you can express the isolated toxins given that they inhibit translation? This is not possible for the majority of TA toxins.

L. 134. Please include a gel showing the stoichiometry as this cannot be seen from the SEC alone (see above).

L. 145. MS measures molecular mass, not weight.

L. 162. The last statement is very blunt - I don't think you can exclude all transcriptional regulation based on this single experiment.

L. 167. Consider if "either" should be replaced by "combinations of".

Fig. 2a and b could be combined into a single panel as the order of the experiments is the same.

Fig. 3a. Consider colouring the monomer in a more informative way, e.g. to show domains, secondary structure etc. The all-purple cartoon doesn't say much. Alternatively, 3a can be left out entirely as all of this can be seen from 3b.

Fig. 3d. Please include the electrostatic potential scale (in $k_B T e^{-1} c$ or $R T e^{-1} c$) from red to blue from APBS.

Fig. 4f. There is no SEM on the last (orange) bar. If the errors are that small, is the difference between red and orange really not significant?

L. 321. What does Mr mean?

Ref: Nature Communications manuscript NCOMMS-24-12797-T

Title: Inducible auto-phosphorylation regulates a widespread family of nucleotidyltransferase toxins

Response to Editor and Reviewers

We would like to thank the Editor and Reviewers for their positive comments and constructive criticism on our revised manuscript. Point-by-point responses to Editorial and Reviewer comments are listed below in blue, and where necessary, changes have been made to the original manuscript and are shown in red. In some instances, changes have been made to figures but these are not altered in color, in order to maintain consistency within the figure.

Reviewer #1 (Remarks to the Author):

Toxin/antitoxin (TA) systems are ubiquitous and diverse regulatory elements that bacteria use to correspond to different environmental stress conditions and in plasmid and bacteriophage defense. The toxins are typically blocking transcriptional and translational processes resulting in reduced energy consumption, cell division and may induce dormancy and allow, for example, bacteria to escape antibiotic treatment. The partner antitoxin typically prevents unwanted toxin activity under 'normal' conditions, typically by forming a stable inactive complex (sequestration) or modifying the toxin. This particular work deals with the nucleotidyltransferase (NTase) family of toxins that block bacterial translation by tRNA processing. *M. tuberculosis* carries 4 gene pairs coding a MenAT system, wherein the toxin MenT is a NTase. Recent work of the author's group provided structural insights in the activity of the toxin NTase, but how the antitoxin MenA counteracts the toxin activity was not yet understood. Curiously, while the 4 MenT genes are largely similar, the antitoxins are more diverse.

In this manuscript, Arrowsmith present an entirely new mechanism by which MenA1 and MenA3 induce autophosphorylation of the toxin. They present convincing biochemical, structural biological and biophysical experiments together with molecular dynamics approaches to support the insight that binding of the antitoxin provokes subtle movements in the MenT1 and MenT3 active site to present a threonine or serine to the gamma phosphoryl of a bound nucleotide and mediate a phosphotransfer to the amino acid. This phosphorylation provokes an electrostatic repulsion and prevents binding of a substrate nucleotide for the NTase action.

Inhibitory autophosphorylation was previously also described as a mechanism for toxin inhibition, for example for the HipA toxin in *E. coli*. However, in this case, phosphorylation occurs in the absence of the antitoxin and is therefore mechanistically different. As such, this manuscript presents a completely novel mechanism of toxin activity regulation induced upon sequestration by the antitoxin. The authors present new insights in translational control in *M. tuberculosis*, the world most infectious killer according to WHO, that just recently published a report disclosing the highest number of people diagnosed with tuberculosis since monitoring started in 1995. Fundamental

studies to unravel molecular processes in *M. tuberculosis* are thus highly demanded. Technically, the manuscript is excellent. The data provided are highly supporting the proposed mechanism and are presented nicely.

We thank the Reviewer for their positive and thorough assessment.

There are a few, rather minor comments to be addressed.

On line 183, a number of mutations are listed that are demonstrated to abolish the phosphorylation. However, the authors do not provide a rationale for the choice of these mutants. There is no reference to previous work on these mutations, for example in the introduction. If the mutants are based on the structural biology work provided in this manuscript, this should be more clearly addressed.

1) We have clarified the selection of conserved active site mutants based on previous functional studies showing essentiality of these residues for NTase activity, including relevant references in the text:

'Next, we assessed the impact of mutations to highly conserved NTase fold residues T39, D41, K137, and D152 on phosphorylation activity, several of which were previously shown to be essential to NTase activity.'

On page 9, LC-MS data are used to demonstrate that T39 is phosphorylated. The supplementary figure indicates that this is most likely based on analysing a tryptic digest peptide mixture. This is not clearly mentioned, nor is this included in the method section.

2) We have adjusted the methods section (Mass Spectrometry) for additional clarity regarding LC-MS/MS analyses.

On page 17, line 366-373, the authors discuss the movements of some residues. The authors should be more consequent in addressing whether this refers to MenT1beta for clarity.

3) We have adjusted the text for additional clarity, ensuring MenT1 beta is referenced when discussing movement of residues:

'When bound to MenT₁β, D142 and C104 form hydrogen bonding interactions with the adenine N-6 and ribose 2' OH respectively, whilst R84 and R40 directly interact with the β,γ-bridging oxygen (Fig. 6D). We concluded that MenA₁ asymmetrical binding has forced movement of the MenT₁β T39 loop towards the bound ATP, facilitating auto-phosphorylation of MenT₁β. In this model, MenT₁β D41 also lies 3.7 Å from T39...'

The supplementary movie is not entirely clear, it may require some extra information. As it is presented now, it starts with a dissociation event followed by a reassociation, while the reverse is probably meant to be presented.

4) The movie (from a well-tempered metadynamics simulation) has been re-rendered, following the suggestion, with an updated figure legend:

‘Well-tempered metadynamics simulation showing dissociation of the MenT₁α-p:MenA₁:MenT₁β-p heterotrimer following phosphorylation of both toxin protomers, and the trend towards dissociation into a MenA₁:MenT₁α-p heterodimer and MenT₁β-p monomer. Structures are shown as cartoons colored dark grey (MenA₁), blue (MenT₁α-p), and yellow (MenT₁β-p). The animation is aligned with the data shown in **Fig. 6H** and **Supplementary Fig. 9D**. The distance between the Centre of Mass (COM) of MenT₁β-p and the COM of MenA₁ + MenT₁α-p was selected as the collective variable (CV).’

Lastly, in the discussion on page 22, line 518, reference is made to the relation of altered ATP levels and toxin (in)activation. However, autophosphorylation of MenT1 also occurs in the presence of ADP which raises question on the ATP concentration dependency for this particular toxin. This deserves to be discussed.

5) We have included additional discussion points with regards to the influence of reduced ATP levels on toxin regulation, with added focus on the inability of ADP to generate sufficiently high levels of phosphorylated toxin:

‘Under such conditions, the requirement for activated toxin to attenuate bacterial growth would be favored by depleted pools of cellular ATP, with the inability of ADP to generate significant levels of phosphorylated toxin supporting the hypothesis that a surplus of NTPs are required for inactivation of the toxin *in vivo*.’

Finally, the manuscript may miss some broader context. The MenT toxins belong to a large family of NTases that is said to be widespread throughout microbial life (as mentioned in the introduction). How about MenA1 and MenA3, is the presented mechanism for MenAT1 or MenAT3 indeed as general as mentioned on line 532 ?

6) We have added an additional experiment (**Fig. 8**) to demonstrate translatability from our predicted models to an *in vivo* setting for MenAT₃, which we feel provides additional weight to the conclusions made regarding the proposed mechanism of auto-phosphorylation. We have also expanded the discussion to propose systems from other organisms in which this mechanism may be utilized as a means of toxin regulation:

‘For example, DUF2253 and DUF4849 proteins are both members of the DUF1814 superfamily that have been proposed to function as putative NTases³⁹, both of which are typically encoded immediately upstream of predicted transcriptional regulators⁸. Analysis of the Conserved Domain Database⁴⁰ revealed strict conservation of a serine or threonine at the site of MenT phosphorylation in all retrieved hits for either COG protein family. One such hit, PygT, encodes a DUF2253 domain and belongs to the functional PygAT TA system from *Pyrococcus yayanosii*. Whilst the toxin has been shown to be activated in response to high hydrostatic pressure⁴¹, the mechanism by which its activity is counteracted remains unknown. Collectively, our findings may help shed light onto the exact mechanisms by which functionally active COG5340 and COG4861 antitoxins negate DUF1814 toxin activity across bacterial and archaeal genomes alike.’

Reviewer #2 (Remarks to the Author):

The paper by Arrowsmith et al., addresses the mechanism of neutralisation of MenT tuberculosis toxins by sequestration and autophosphorylation. The authors used a combination of X-ray crystallography and MD, biochemistry and biophysics with classical microbiology to investigate the regulation of various types of toxins that are controlled by very different antitoxins. This is a solid and well written manuscript that deconstructs different aspects of the regulation of these toxins to extract meaningful insights into their mechanism of action and neutralisation.

We are grateful for this positive summary, thank you.

The one major concern I have regarding their conclusions relates to their use of a T39A substitution and the interpretation of the results involving these results. The authors performed several measurements with T39A including experiments aimed at qualifying the possible role of T39 in antitoxicity. The experiments of Fig 5D are used to claim that T39 is not essential for antitoxicity based on the observation that the toxin retains activity. However one can also interpret this as possible effects in the catalytic cycle of the enzyme introduced by the substitution of Thr for Ala (while tRNA is still slightly modified, the activity of the enzyme appears severely affected). To support these claims the authors should use for example a Cys substitution which is closer to Thr both in shape and properties.

1) We would like to clarify the experiment in **Fig. 5D**, which shows that (i) T39 is not essential for *toxicity* due to the observed continued modification of tRNA, as also seen in **Fig. 5C** and in previous data demonstrating toxicity *in vivo* (Xu et al 2023 *Nat comms* Fig. S6; <https://www.nature.com/articles/s41467-023-40264-3>); and (ii) T39 phosphorylation is not the only requirement for antitoxicity, as incubating MenA₁ with T39A to examine binding without phosphorylation still reduces the levels of tRNA modification by the T39A active mutant. This allows us to separate the impacts of antitoxin binding and complex formation, from the impact of phosphorylation.

These results do show a reduction in NTase activity with the T39A mutant, and this matches a previously observed reduction, but not abolition of, toxicity *in vivo* (Xu et al 2023 *Nat comms*). We agree it is sensible to consider a different mutation at this position and so as requested, we also made a T39C mutant and tested this in *in vivo* toxicity experiments, now presented as **Fig. 5E** and discussed in the accompanying text. The T39C had the same levels of toxicity as the T39A mutant, and both mutants could still be neutralized *in vivo*. This suggests, as the Reviewer has kindly pointed out, that T39 does have a role for full NTase activity, but is not essential. This is now commented upon in the manuscript within lines 323-330.

Analytical SEC is described in the Methods as done in a 1030 column, however the experiments from Fig. 1B are clearly from a bigger column. Could the authors please comment?

2) SEC traces in **Fig. 1B** represent the final stage of preparative co-expressions as detailed in the methods (Protein Purification), and the column used is stated in the figure legend. We

have also added the name of the analytical column used for **Fig. 5F-H** to the legend, for additional clarity.

Related with this, the colours and representations from 1B make it difficult to interpret the figure especially the middle panel. In addition, the use of Absorption vs MW axes used in Fig. 5 is not a usual way of representing SEC. For example is difficult to assess from this figure the void volume of the column.

3) SEC traces in **Figs. 1, 5, and Supplementary Fig. 8** have been amended accordingly to display absorbance vs elution volume, ensuring traces are more visible by using darker colors.

It would be helpful to show T39, D41 and K137 in Fig. 2.

4) We have shown these residues as requested.

It would be good in Fig. 3, instead of the two views of the toxin to have a second a panel with a zoom to the TPO showing an omit map of the phospho-threonine.

5) We have moved the 2Fo-Fc map from **Supplementary Fig. 5D** to **Fig. 3B**.

How conserved is T39 (or a possible Ser substitution) at this position?

6) This was an interesting point to explore; we have added discussion points (~line 527) with regards to the high conservation of Ser/Thr at this position, with reference to existing literature documenting conservation of DUF1814 motifs:

'For example, DUF2253 and DUF4849 proteins are both members of the DUF1814 superfamily that have been proposed to function as putative NTases, both of which are typically encoded immediately upstream of predicted transcriptional regulators. Analysis of the Conserved Domain Database revealed strict conservation of a serine or threonine at the site of MenT phosphorylation in all retrieved hits for either COG protein family.'

Is there a reason why the oxygen atoms of phosphate groups are not shown?

7) Our apologies, figures have been amended to show oxygen atoms of phosphate groups.

The authors should provide the AlphaFold score of the models reported in the paper.

8) Predicted template modelling (pTM) scores for models referenced have been added to the manuscript in the respective positions and are as follows:

MenA₁:MenT₁ heterodimer – 0.89 / MenA₃ – 0.87 / MenA₃:MenT₃ heterodimer – 0.84

While the docking experiments are an important part of the paper, they are poorly described in the Methods section.

- a) How was the active site defined?
b) Can they plot the differences (in score, rmsd) between the the selected poses and the rest?

9) An additional section (Molecular Docking; see line 874) has been added to the Materials and Methods to provide additional information regarding the protocol, the definition of the binding site and validation of the poses selected for the MD simulations. Please see additional panels **Supplementary Fig. 9H-J**.

Reviewer #3 (Remarks to the Author):

In this work, the authors have studied nucleotidyltransferases and how in *M. tuberculosis* auto-phosphorylation of the nucleotidyltransferases can provide a regulatory function through modifying the toxin—anti-toxin interplay that is so crucial to the life cycle of the bacteria. In particular, they focus on nucleotidyltransferases in *Mtb* that are toxins which can be controlled by their corresponding antitoxins. The work is worth publishing, after the following concerns are addressed.

Thank you for this assessment.

Major concerns.

“MenT toxins can auto-phosphorylate to control activity, catalysed by antitoxin-dependent movement of the target active site residue towards the donor phosphate.” The calculations the authors have done cannot conclusively show this. Such results should be couched as “predictions” or “calculated results”.

1) The manuscript has been amended to clarify these results are predictions based on simulations. In the abstract it now reads, ‘Finally, we expand this **predicted** model to...’ and within the last paragraph of the introduction we say, ‘Our **calculated results suggest** that MenT toxins can auto-phosphorylate...’, then at the start of the discussion we say, ‘We **predict** that neutralization ...’.

Also, the authors should report in more detail on their analysis of the motion of the active site residues.

2) We kindly direct the Reviewer to point 6) in response to Reviewer #4, in which we have expanded our analysis into the movement of the T39 loop (see lines 233-235).

The docking protocols utilized have not been explained. Instead, we are told that the structures were prepared for docking and then are told the results. That is not sufficient information for reproducibility. The choices made for docking can affect whether the results are sensible or not, so therefore it is not possible for the reader to judge the suitability of the docking methods without that information.

3) An additional section (Molecular Docking; see line 874) has been added to the Materials and Methods to provide additional information regarding the protocol, the definition of the binding site and validation of the poses selected for the MD simulations. Please see additional panels **Supplementary Fig. 9H-J**.

Fig. S9b. shows that changes are still occurring after 300 ns. Significant changes are happening even in the last 50 ns. It is better to run the simulations longer. Also, it is best not to rely on a single simulation for each set of conditions but to have at least triplicate simulations. The authors mention having run simulations in triplicate but then do not show their results. They should show the results in the supporting information, so that the readers can see how representative are the simulations presented in the main text. I.807. The equilibration is too short. Make it at least twice as long, based on standard practice of how much equilibration is needed for such a system.

4) As recommended, simulations were repeated over 500 ns and with longer equilibrations. We did not observe any additional changes as a result. **Supplementary Fig. 9A-C** have been re-plotted to show mean values, with standard deviation of triplicate data now also shown as faded lines.

Figure 1 illustrates major problems with the MS data reporting. That also affects several of the Supporting images. See the list of concerns under Fig. S1, below. For Fig. 1c:

--Axes should be labeled, a standard requirement for scientific publications. What quantity is presented on the y-axes of the charts? "percent" or "Miller units" refer to the values along the scale but: percent of what? Meanwhile, for the x-axis, "mass" is the quantity but the units should be specified or an expression of mass-to-charge ratio should be given in parentheses (M/Z).

--Notice that the signal-to-noise ratio is much better for the 2nd chart than for the first or third chart. The authors should provide an explanation about that. Also, there is a strange offset for the charts, such that the baseline is above 0. That simply looks like sloppy positioning of the axes, but may indicate a slight initial percent rise at the start of the experiment.

5) All graphs have been reformatted to address the issues raised, including adding appropriate axes titles, and re-processing spectra to ensure baselines are normalised and to standardise signal-to-noise ratios.

Fig. 1.

(d) There are two images which represent separate experiments but they are labeled identically. Instead, their overall label should be different (to match what is different between the experiments, as described in the caption).

6) The figures differ in their ATc labels at the 12 o'clock positions, which we have enlarged for clarity. We have also amended the figure legend to be more clear with regards to the role of ATc. Please note this panel has been moved to **Supplementary Fig. 1F**, and replaced with the liquid growth quantifications of LacZ activity that were originally in the **Supplementary Materials**, as we felt the quantitative data are more impactful having now been enhanced with the addition of a positive control and therefore should be in the main figures.

Fig.3.

--"orange for phosphate". Do the authors mean "orange for phosphorus"? Or is the orange stick supposed to represent the whole steric bulk of a phosphate group? In the latter case, it would be best to show the P and all its attached O's, not just a single stick, so that its role to provide steric

bulk is more obvious from the image. As II.224-225 state, the phosphate provides “steric occlusion of the active site”, but that does not appear clear in the current version of the figure.

7) We have changed the legend to ‘orange for phosphorus’, and shown individual phosphates to illustrate the steric impacts of phosphorylation.

L.403. Methodological choices utilized should be specified for the two computational methods mentioned in this sentence.

8) An additional section (Molecular Docking; see line 874) has been added to the Materials and Methods to provide additional information regarding the protocol, the definition of the binding site, and validation of the poses selected for the MD simulations.

II.606-607. Make sure to label properly concentration (in molar units, such as “nanomolar” or “nM”) and quantity (number of particles, such as “nanomoles”).

9) We now state the volume mixed together, and had already stated the number of particles in nmol.

I.795. “poses were subsequently filtered based on ... desirable physicochemical properties”. Docking poses should not be ruled out based on desirable pchem properties. What properties were used? Why would that be a sensible approach to distinguish between poses for one particular molecule, considering that a molecule’s physicochemical properties hold true irrespective of their docking pose?

10) We thank the Reviewer for their insight here – we have re-processed the docking poses without filtering based on physicochemical properties. This had no effect on the best-scored poses.

I.796. (also II.272-273.) What is a “highest affinity” pose? Docking yields docking scores which are not the same as binding affinities. It is best to use the term “best-scored poses”.

11) We have reworded to ‘best-scored’ where appropriate.

I.796. Was the filtering done before or after the best docked pose was selected?

12) Filtering was performed before selection of the best-scored pose - we have re-processed the docking poses and omitted filtering based on physicochemical properties, which had no effect on the best-scored poses. The manuscript has been adjusted accordingly: ‘500 poses were generated for each ligand, with poses subsequently filtered based on steric clashes and torsion. Best-scored poses for each ligand were selected for subsequent analysis and comparison.’

Fig S1.

--“+/-“ should be used instead of “-/+”

--Axes should be labeled, a standard requirement for scientific publications. What quantity is presented on the y-axis of the charts? "percent" or "Miller units" refer to the values along the scale but: percent of what? Meanwhile, for the x-axis, "mass" is the quantity but the units should be specified or an expression of mass-to-charge ratio should be given in parentheses (M/Z).

--Notice that the signal-to-noise ratio is much better for charts (b) and (c) compared to (a). The authors should provide an explanation about that. Also, there is a strange offset for (c), such that the baseline is above 0. That simply looks like sloppy positioning of the axes.

--The first lane in figure (d) looks unnecessary. Is it blank? Just 0? It appears empty. If it is important, add this to the caption. In any case, do not put the legend right above that lane, but have it centered horizontally instead, to avoid confusion.

--The lanes in (d) should be labeled more completely; note that the figure caption points out some aspects of the lane contents that are not obvious from the lane label in the chart. So for example in the last lane it is not only P + MenAT1 but also pGMC and pJEM15, right? Those extra items should be added to the lane label or else the lane label is confusing. It might be better to reduce the lane label to a simpler form (such as a number) and then make sure the caption itself is as clear as possible.

13) The MS plots have been altered as requested. We have also moved the legend to the right of the graph, and have clarified the contents of each lane in the figure legend.

Fig S2.

--For Fig. S2, also, it would be better to label each panel with a letter, instead of only labeling the three columns. Some of the items in column a do not correspond to what is found in columns b and c. So it can be very confusing to a reader who is trying quickly to understand what is in the figure and how things are arranged.

--Notice that the signal-to-noise ratio is much better for almost all the charts than for the third one in section (a). The authors should provide an explanation about that.

--It is not sufficient to say that each experiment was done three times and a representative image is shown. Explain more. There is lots of confusion in the literature when triplicates are mentioned but exactly what was done is not explained well. Was the MS of the same sample run three times or were all the experimental procedures repeated three times? What is the technical definition of "representative"? Quantify that.

14) We have added letters to each panel for additional clarity, normalised as requested previously, and amended the figure legend to further clarify that each spectra represents three independent biological replicates.

Under (c), three of the charts have another rather tall peak just a little higher than the main peak. Explain.

15) Following normalisation we cannot spot the peaks in question, and conclude they would not have represented an abundant species of note.

Figs S3-S5, S7. See comments on early supporting figures.

16) We have amended formatting and labelling accordingly in line with previous comments.

Fig S6.

--For (b), "data are representative of two independent experiments"; in what way are they representative? After that it is stated "bars display mean +/- SEM". So was the experiment done twice but each time in triplicate? Very confusing.

--The caption has the presence of MenA1 ascribed backwards to charts (c) and (d).

17) We have clarified that each experiment consists of independent biological replicates, and amended the figure legend accordingly.

Fig. S7.

--Are those best docking poses or did you impose a requirement that the LHS be superimposed? Explain more about the rank of the chosen poses for the different molecules.

--What is a "highest affinity" pose? Docking yields docking scores which are not the same as binding affinities. It is best to use the term "best-scored poses".

--"Superposition of poses" can have two meanings. In one case, a computational procedure is used to align poses. In the other meaning, images are unchanged but are overlaid. Most likely the authors simply overlaid multiple docked structures without superposition. So the term should be an "overlay of poses". If instead the authors did an alignment, they should justify why they did that and what relevance that is to the scientific inquiry.

18) Each pose represents the best-scored docking pose for each molecule. We have amended phrasing to clarify that poses are 'best-scored' rather than 'highest-affinity' and further clarified that poses are structurally overlaid onto one another, and not aligned computationally.

Fig. S8.

--"greater than 10 less than or equal to 20" is technically incorrectly written. Could be fixed by adding "and" in the middle ("greater than 10 and less than or equal to 20") or could be written in a more compact form.

--Not "R₂" but "R²" (superscript).

19) The figure legend has been amended accordingly.

Reviewer #4 (Remarks to the Author):

Overall, the manuscript by Arrowsmith et al. presents a convincing and highly intriguing account of how antitoxin binding induces phosphorylation in a widespread family of NTase toxins and

repurposes the toxin active site from nucleotide transfer towards auto-phosphorylation in order to control activity. The study elegantly combines in vitro functional studies using a wide range of biochemical assays with structure prediction and experimental structure determination as well as molecular modelling to arrive at a cohesive and convincing molecular model. Overall, the manuscript is highly interesting and is suitable for publication following careful revision. However, in the current version, several of the figure panels are somewhat preliminary and could be improved. My main concerns therefore relate to presentation of the data as outlined below.

Thank you for these positive comments.

Several of the figure legends are too brief and do not provide the required information to interpret the data, e.g. 1a, 1d and S5c, see specific comments below. Please carefully check that all figure elements are described in all legends including all lanes/bars/labels.

1) We thank the Reviewer for this feedback. Figure legends have been checked to ensure the information provided allows the reader to fully interpret the data.

L. 180. I am not convinced that you can conclude that Mg²⁺ is not required. Have you considered that it could be carried along by one of the protein or nucleotide components, consistent with the observation that you see increased activity with added Mg²⁺? In fact, I would be surprised if Mg²⁺ was not required.

2) We thank the Reviewer for their insight here. We repeated phosphorylation experiments by first pre-incubating protein components with EDTA for 1 hour, then buffer exchanged to remove EDTA. CTP was then added to samples, either in the absence or presence of MgCl₂. In the absence of MgCl₂, no phosphorylation could be detected, confirming Mg is indeed essential for phosphorylation:

‘Having identified increased phosphorylation activity in the presence of MgCl₂, we sought to establish whether magnesium was essential for MenT₁ phosphorylation. MenT₁ and MenA₁ were first pre-incubated with 5 mM EDTA for 1 hour to facilitate chelation of protein-bound metals prior to overnight dialysis. Supplementation of protein mixtures with 1 mM CTP alone failed to induce a change in the mass of MenT₁ (**Supplementary Fig. 2I**). In contrast, supplementation with 1 mM CTP and 10 mM MgCl₂ produced an increase in mass of 81 Da corresponding to the addition of a phosphate (**Supplementary Fig. 2J**), confirming magnesium is essential for phosphorylation activity.’

For bar charts, I believe it is recommended to all measured data points as individual markers if n<10, not just mean +/-SEM. Since you do not have that many repetitions, I suggest you follow that standard for all bar charts.

3) All individual points have been shown in line with standard practice for data where n<10.

Overall, I am disappointed with the quality of the SDS-PAGE (PhosTag) gels, most of which have run very unevenly. Sometimes, like in Fig. 2a and S6a this even makes it hard to interpret the data. What is the reason for this? We've done Phos-tag gels and I believe they are regular SDS-PAGE gels with an extra reagent added, so it should be straightforward to make them run more smoothly with commercial gels. Consider repeating some of the most important ones (like those in the main figures) for a better and more convincing overall presentation of the paper.

4) We argue that the gels can still be readily interpreted and that the conclusions are supported by our MS data. We did try to re-run these experiments, but have found it very hard to procure additional gels. Whilst we accept this is not the best response, having ordered additional gels three months ago to repeat these experiments, they still have not arrived from Japan. We have decided to keep the gel images we have.

Figure 2f. How can the mean of a integrated band intensities be zero with a large SEM (WT + bar)? An integrated band intensity cannot be negative, so if the SEM is large (about +/- 20% here) then the mean must be positive and >0. Please check the data once more.

5) We thank the Reviewer for spotting this mistake; this dataset had been incorrectly normalised. Data now reflects raw density, which matches all other lanes.

L. 223. For the RMSD, it appears that all atoms were used (1127)? I don't think this the best way to reveal folding changes (side chains can be floppy and are affected by crystal contacts). Instead, I would calculate RMSD using C-alpha atoms only and compare the values for the entire protomer to that calculated from the loop region only to show that the loop moves upon phosphorylation.

6) Alignments were repeated to show a) sequence-independent overlay of entire protomers, b) alignment of C-alpha atoms across entire protomers, and c) alignment of C-alpha atoms across the loop region from either protomer:

'Sequence-independent structural overlay of MenT₁ and MenT₁-p returned a root mean square deviation (RMSD) of 0.446 Å (1127 atoms), indicating a near identical topology of core domains and folds (Fig. 3C). Alignment of each structure via C-alpha atoms only, returned an RMSD of 0.370 Å (1127 atoms), whereas aligning the C-alpha atoms of just the respective T39 loops returned an RMSD of 0.778 Å (7 atoms), indicating a larger shift between these specific loops.'

L. 248. Isn't the difference in melting point just 1.4° between CTP and the other nucleotides? From the figures, it appears that the thermal shift is 3.8° for CTP and 2.4° for the others, so a 1.4° difference between them, please check this.

7) We thank the Reviewer for spotting the error, we have adjusted the text accordingly to rectify this mistake.

L. 272. Is there no structure of a homologous protein with nucleotide bound that you can use for comparison of the binding mode? It would be nice to confirm the docking poses. Also, I find it very

odd that you observe no base-specific interactions given that the NTase prefers CTP. Can you comment on this?

8) Whilst the manuscript was in review, another group published the crystal structure of MenT3 bound to CTP (<https://doi.org/10.1093/nar/gkae177>). We have compared the MenT3 ATP MD pose to that of the solved structure for reference, and observed a close match (**Supplementary Fig. 9E-F**, lines 452-465). With regards to the MenT1 NTP docking poses, we have added an additional reference to a related NTase, Cid1, which forms the majority of its protein-ligand contacts through interactions with the triphosphate tail of each NTP. Finally, we feel that scrutinization of differences in NTP binding modes for a structural explanation for the observed preference of MenT1 for CTP as an NTase substrate does not fit with the scope of our research, as the toxin exhibits no preference for phosphorylation substrates, which our docking models support. It is likely we will need to also consider the tRNA target to help analyze CTP preference for the NTase activity, in a future study.

Fig. 5efg. I don't believe it's appropriate to have MW on the x-axis of a SEC trace. You should always have elution volume (the true variable) and then indicate with arrows the elution volumes and MW your standard proteins. Also, it looks like MenT-p alone shifts compared to the unphosphorylated versions, can you comment on this?

9) SEC traces have been amended accordingly. With regards to MenT1-p elution, we note that the calculated R_{st} value of MenT1-p from its crystal structure is smaller than for MenT1 (see **Supplementary Fig. 8**), which may explain the difference in the elution trace for MenT1-p relative to MenT1 WT/T39A.

Fig. S5d. Please include the unbiased Fo-Fc difference map from before modelling the phosphate group into the structure. Refined 2Fo-Fc is biased by the model and could in principle show artificial phosphate group density.

10) We have included the unbiased difference map to reflect the presence of missing density prior to modelling of the phosphate (see **Supplementary Fig. 5E**).

Figure 1a. It is not clear from the figure if it is the N-terminal or C-terminal parts of the toxins that are conserved. I would suggest that you use dashed lines going across the isoforms or similar colours of domains to indicate conserved parts now that the figure is drawn to scale anyway. In the legend1, what does "original and revised nomenclature" refer to?

11) As a visual aid, we have aligned each of the four MenT toxins to illustrate conserved motifs, see Fig. S1a. We kindly direct the Reviewer to Dy et al 2014 *NAR* Fig. 8a, in which the authors illustrate the four conserved DUF1814 motifs and where they are located: <https://academic.oup.com/nar/article/42/7/4590/2436634?login=true>. We have also clarified in the figure legend that original and revised nomenclature refers to original gene identifiers and revised nomenclature of gene products.

Figure 1b. I would be more cautious about listing the estimated MW for the SEC experiments. SEC elution volumes depend on hydrodynamic radii, which are only app. corresponding to MW, so it makes no sense to list MW so precisely (5 digits). Also, please include a gel showing whether (or not) the two proteins bind and co-elute with a tag on one protein in the three cases.

12) Values shown were calculated (known) MW values for respective proteins, however, for the sake of clarity as this was not explicitly stated, SEC traces have been adjusted accordingly to simply show elution volumes for each sample. We refer the Reviewer to Cai et al 2020 *Sci Adv* Fig. S4 for our previously published co-elution gels <https://www.science.org/doi/10.1126/sciadv.abb6651>.

Figure 1d. What does ATc +/- refer to? Also, the blue colour is very faint, so I would like to ask you to include a positive control where there is a clear repression of the lac promoter.

13) The figure legend has been updated for additional clarity with regards to the use of the pGMC inducer anhydrotetracycline (ATc). As requested, we included a positive control in β -galactosidase assays to show clear repression of the *parDE1* promoter by the ParDE1 complex. Please also note this panel has been moved to **Supplementary Fig. 1F**, and replaced with the liquid growth quantifications of LacZ activity that were originally in the Supplementary Materials, as we felt the quantitative data are more impactful having now been enhanced with the addition of a positive control and therefore should be in the main figures.

'We examined whether there was **also** any transcriptional regulation by cloning the 1000 bp region immediately upstream of the *menAT1* transcriptional start site into the promoterless *lacZ* fusion construct pJEM15²⁸, and quantified β -galactosidase activity in *M. smegmatis* during co-induction of MenA₁, MenT₁, or both together (**Fig. 1D; Supplementary Fig. 1F**), **using ParDE1 as a positive control for transcriptional repression.'**

L. 132. It should be mentioned how you can express the isolated toxins given that they inhibit translation? This is not possible for the majority of TA toxins.

14) We have added additional detail to lines 134-137:

'Despite demonstrable toxicity of MenT NTases high yields of each lone toxin homologue (2-5 mg/L) can be obtained following recombinant protein expression in *E. coli* using rich media, which we previously suggested may be a result of elevated tRNA target levels in *E. coli* relative to *M. tuberculosis*.'

L. 134. Please include a gel showing the stoichiometry as this cannot be seen from the SEC alone (see above).

15) We have amended **Supplementary Fig. 1** to include calculated/observed Molecular Weight and Stokes radius ratios for MenT1 lone and co-expression samples to reflect that the co-expression peak matches the expected Rst of the heterotrimer.

L. 145. MS measures molecular mass, not weight.

16) We have amended the text accordingly and thank the Reviewer for spotting this mistake.

L. 162. The last statement is very blunt - I don't think you can exclude all transcriptional regulation based on this single experiment.

17) We have altered our concluding statement to be more conservative.

L. 167. Consider if "either" should be replaced by "combinations of".

18) We have amended the text accordingly.

Fig. 2a and b could be combined into a single panel as the order of the experiments is the same.

19) Both panels are qualitative (a) and quantitative (b) analyses of the same experiment and are split into separate panels for clarity in line with other figures.

Fig. 3a. Consider colouring the monomer in a more informative way, e.g. to show domains, secondary structure etc. The all-purple cartoon doesn't say much. Alternatively, 3a can be left out entirely as all of this can be seen from 3b.

20) The colouring of the toxin has been altered to depict secondary structure elements.

Fig. 3d. Please include the electrostatic potential scale (in $k_B T e^{-1} c$ or $R T e^{-1} c$) from red to blue from APBS.

21) The requested potential scale has been included for reference.

Fig. 4f. There is no SEM on the last (orange) bar. If the errors are that small, is the difference between red and orange really not significant?

22) Adjusted p-values were determined to be 0.9997, 0.9954, and 0.4964 when comparing changes in T_m when incubated with UTP vs ATP, CTP, and GTP, respectively. As such, no statistical significance in the changes to T_m were detected.

L. 321. What does Mr mean?

23) Mr referred to molecular weight; we have amended this for clarity.

Reviewers' Comments:

Reviewer #1:

Remarks to the Author:

The authors have adequately responded to the remarks and suggestion provided during the first review.

Reviewer #2:

Remarks to the Author:

The authors have addressed all the concerns and suggestions that I had made in the original review. I am satisfied with the response. I would like to congratulate the authors on a nice discovery and interesting findings.

Reviewer #4:

Remarks to the Author:

See attached document with colours.

Reviewer #4 (Remarks to the Author):

Overall, the manuscript by Arrowsmith et al. presents a convincing and highly intriguing account of

how antitoxin binding induces phosphorylation in a widespread family of NTase toxins and repurposes the toxin active site from nucleotide transfer towards auto-phosphorylation in order to control activity. The study elegantly combines in vitro functional studies using a wide range of biochemical assays with structure prediction and experimental structure determination as well as molecular modelling to arrive at a cohesive and convincing molecular model. Overall, the manuscript is highly interesting and is suitable for publication following careful revision. However, in the current version, several of the figure panels are somewhat preliminary and could be improved. My main concerns therefore relate to presentation of the data as outlined below.

Thank you for these positive comments.

Several of the figure legends are too brief and do not provide the required information to interpret the data, e.g. 1a, 1d and S5c, see specific comments below. Please carefully check that all figure elements are described in all legends including all lanes/bars/labels.

1) We thank the Reviewer for this feedback. Figure legends have been checked to ensure the information provided allows the reader to fully interpret the data.

OK

L. 180. I am not convinced that you can conclude that Mg²⁺ is not required. Have you considered that it could be carried along by one of the protein or nucleotide components, consistent with the observation that you see increased activity with added Mg²⁺? In fact, I would be surprised if Mg²⁺ was not required.

2) We thank the Reviewer for their insight here. We repeated phosphorylation experiments by first pre-incubating protein components with EDTA for 1 hour, then buffer exchanged to remove EDTA. CTP was then added to samples, either in the absence or presence of MgCl₂. In the absence of MgCl₂, no phosphorylation could be detected, confirming Mg is indeed essential for phosphorylation:

‘Having identified increased phosphorylation activity in the presence of MgCl₂, we sought to establish whether magnesium was essential for MenT₁ phosphorylation. MenT₁ and MenA₁ were first pre-incubated with 5 mM EDTA for 1 hour to facilitate chelation of protein-bound metals prior to overnight dialysis. Supplementation of protein mixtures with 1 mM CTP alone failed to induce a change in the mass of MenT₁ (**Supplementary Fig. 2I**). In contrast, supplementation with 1 mM CTP and 10 mM MgCl₂ produced an increase in mass of 81 Da corresponding to the addition of a phosphate (**Supplementary Fig. 2J**), confirming magnesium is essential for phosphorylation activity.’

OK, good to have this clarified.

For bar charts, I believe it is recommended to all measured data points as individual markers if n<10, not just mean +/-SEM. Since you do not have that many repetitions, I suggest you follow that standard for all bar charts.

3) All individual points have been shown in line with standard practice for data where n<10.

OK, this is perhaps more of an esthetic comment, but it's hard to see the red crosses, especially when the bar outline is red as well, like in Fig. 4e and f. I think the bar charts are over-complicated by too many colours that are not needed. For example, you don't need each experiment to be represented by a different colour if there is a label below. Also, I would streamline fonts and their sizes as well as colours and styles more across the figures as currently each new bar chart has a different style from the previous one.

Overall, I am disappointed with the quality of the SDS-PAGE (PhosTag) gels, most of which have run very unevenly. Sometimes, like in Fig. 2a and S6a this even makes it hard to interpret the data. What is the reason for this? We've done Phos-tag gels and I believe they are regular SDS-PAGE gels with an extra reagent added, so it should be straightforward to make them run more smoothly with commercial gels. Consider repeating some of the most important ones (like those in the main figures) for a better and more convincing overall presentation of the paper.

4) We argue that the gels can still be readily interpreted and that the conclusions are supported by our MS data. We did try to re-run these experiments, but have found it very hard to procure additional gels. Whilst we accept this is not the best response, having ordered additional gels three months ago to repeat these experiments, they still have not arrived from Japan. We have decided to keep the gel images we have.

I agree that this is not the best response and I think the gel quality detracts from the strength of the conclusions drawn from Fig. 2a and 2e. Given that you have two data points, could you include the other gels in Supplementary Information for transparency? I think it's a bit problematic to integrate gels of this quality, especially for the L14R/V19R mutant, in which it's hard to see the phosphorylated protein band.

Figure 2f. How can the mean of a integrated band intensities be zero with a large SEM (WT + bar)? An integrated band intensity cannot be negative, so if the SEM is large (about +/- 20% here) then the mean must be positive and >0. Please check the data once more.

5) We thank the Reviewer for spotting this mistake; this dataset had been incorrectly normalised. Data now reflects raw density, which matches all other lanes.

OK, more clear, thanks.

L. 223. For the RMSD, it appears that all atoms were used (1127)? I don't think this the best way to reveal folding changes (side chains can be floppy and are affected by crystal contacts). Instead, I would calculate RMSD using C-alpha atoms only and compare the values for the entire protomer to that calculated from the loop region only to show that the loop moves upon phosphorylation.

6) Alignments were repeated to show a) sequence-independent overlay of entire protomers, b) alignment of C-alpha atoms across entire protomers, and c) alignment of C-alpha atoms across the loop region from either protomer:

'Sequence-independent structural overlay of MenT₁ and MenT₁-p returned a root mean square deviation (RMSD) of 0.446 Å (1127 atoms), indicating a near identical topology of core domains and folds (Fig. 3C). Alignment of each structure via C-alpha atoms only, returned an RMSD of 0.370 Å (1127 atoms), whereas aligning the C-alpha atoms of just the respective T39 loops returned an RMSD of 0.778 Å (7 atoms), indicating a larger shift between these specific loops.'

Not sure why you need the entire promoter overlay if the point is just that the fold is the same? Also, in the text above, how can you have the same number of atoms (1127) for both all-atom and Calpha-only overlays?

L. 248. Isn't the difference in melting point just 1.4° between CTP and the other nucleotides? From the figures, it appears that the thermal shift is 3.8° for CTP and 2.4° for the others, so a 1.4° difference between them, please check this.

7) We thank the Reviewer for spotting the error, we have adjusted the text accordingly to rectify this mistake.

OK

L. 272. Is there no structure of a homologous protein with nucleotide bound that you can use for comparison of the binding mode? It would be nice to confirm the docking poses. Also, I find it very odd that you observe no base-specific interactions given that the NTase prefers CTP. Can you comment on this?

8) Whilst the manuscript was in review, another group published the crystal structure of MenT3 bound to CTP (<https://doi.org/10.1093/nar/gkae177>). We have compared the MenT3 ATP MD pose to that of the solved structure for reference, and observed a close match (**Supplementary Fig. 9E-F**, lines 452-465). With regards to the MenT1 NTP docking poses, we have added an additional reference to a related NTase, Cid1, which forms the majority of its protein-ligand contacts through interactions with the triphosphate tail of each NTP. Finally, we feel that scrutinization of differences in NTP binding modes for a structural explanation for the observed preference of MenT1 for CTP as an NTase substrate does not fit with the scope of our research, as the toxin exhibits no preference for phosphorylation substrates, which our docking models support. It is likely we will need to also consider the tRNA target to help analyze CTP preference for the NTase activity, in a future study.

OK

Fig. 5efg. I don't believe it's appropriate to have MW on the x-axis of a SEC trace. You should always have elution volume (the true variable) and then indicate with arrows the elution volumes and MW your standard proteins. Also, it looks like MenT-p alone shifts compared to the unphosphorylated versions, can you comment on this?

9) SEC traces have been amended accordingly. With regards to MenT1-p elution, we note that the calculated R_{st} value of MenT1-p from its crystal structure is smaller than for MenT1 (see **Supplementary Fig. 8**), which may explain the difference in the elution trace for MenT1-p relative to MenT1 WT/T39A.

OK, but if you have done MW standards for the SEC column you could put those as points on the x axis (with an arrow pointing to their elution volume) to let the reader assess the apparent mass of your complexes.

Fig. S5d. Please include the unbiased Fo-Fc difference map from before modelling the phosphate group into the structure. Refined 2Fo-Fc is biased by the model and could in principle show artificial phosphate group density.

10) We have included the unbiased difference map to reflect the presence of missing density prior to modelling of the phosphate (see **Supplementary Fig. 5E**).

OK, looks convincing.

Figure 1a. It is not clear from the figure if it is the N-terminal or C-terminal parts of the toxins that are conserved. I would suggest that you use dashed lines going across the isoforms or similar colours of domains to indicate conserved parts now that the figure is drawn to scale anyway. In the legend, what does "original and revised nomenclature" refer to?

11) As a visual aid, we have aligned each of the four MenT toxins to illustrate conserved motifs, see Fig. S1a. We kindly direct the Reviewer to Dy et al 2014 *NAR* Fig. 8a, in which the authors illustrate the four conserved DUF1814 motifs and where they are located: <https://academic.oup.com/nar/article/42/7/4590/2436634?login=true>. We have also clarified in the figure legend that original and revised nomenclature refers to original gene identifiers and revised nomenclature of gene products.

I would still prefer to have Figure 1a show this alignment by having the domain boxes somehow related by vertical lines, but this is my personal taste and I accept that this is at the authors' discretion.

Figure 1b. I would be more cautious about listing the estimated MW for the SEC experiments. SEC elution volumes depend on hydrodynamic radii, which are only app. corresponding to MW, so it makes no sense to list MW so precisely (5 digits). Also, please include a gel showing whether (or not) the two proteins bind and co-elute with a tag on one protein in the three cases.

12) Values shown were calculated (known) MW values for respective proteins, however, for the sake of clarity as this was not explicitly stated, SEC traces have been adjusted accordingly to simply show elution volumes for each sample. We refer the Reviewer to Cai et al 2020 *Sci Adv* Fig. S4 for our previously published co-elution gels <https://www.science.org/doi/10.1126/sciadv.abb6651>.

OK

Figure 1d. What does ATc +/- refer to? Also, the blue colour is very faint, so I would like to ask you to include a positive control where there is a clear repression of the lac promoter.

13) The figure legend has been updated for additional clarity with regards to the use of the pGMC inducer anhydrotetracycline (ATc). As requested, we included a positive control in β -galactosidase assays to show clear repression of the *parDE1* promoter by the ParDE1 complex. Please also note this panel has been moved to **Supplementary Fig. 1F**, and replaced with the liquid growth quantifications of LacZ activity that were originally in the Supplementary Materials, as we felt the quantitative data are more impactful having now been enhanced with the addition of a positive control and therefore should be in the main figures.

'We examined whether there was **also** any transcriptional regulation by cloning the 1000 bp region immediately upstream of the *menAT1* transcriptional start site into the promoterless *lacZ* fusion construct pJEM15²⁸, and quantified β -galactosidase activity in *M. smegmatis* during co-induction of MenA₁, MenT₁, or both together (**Fig. 1D; Supplementary Fig. 1F**), using ParDE1 as a positive control for transcriptional repression.'

OK

L. 132. It should be mentioned how you can express the isolated toxins given that they inhibit translation? This is not possible for the majority of TA toxins.

14) We have added additional detail to lines 134-137:

'Despite demonstrable toxicity of MenT NTases high yields of each lone toxin homologue (2- 5 mg/L) can be obtained following recombinant protein expression in *E. coli* using rich media, which we previously suggested may be a result of elevated tRNA target levels in *E. coli* relative to *M. tuberculosis*.'

OK

L. 134. Please include a gel showing the stoichiometry as this cannot be seen from the SEC alone (see above).

15) We have amended **Supplementary Fig. 1** to include calculated/observed Molecular Weight and Stokes radius ratios for MenT1 lone and co-expression samples to reflect that the co-expression peak matches the expected Rst of the heterotrimer.

I don't think this adds much. I maintain that an SDS-PAGE gel would be very useful to support the proposed stoichiometry.

L. 145. MS measures molecular mass, not weight.

16) We have amended the text accordingly and thank the Reviewer for spotting this mistake.

OK

L. 162. The last statement is very blunt - I don't think you can exclude all transcriptional regulation based on this single experiment.

17) We have altered our concluding statement to be more conservative.

OK

L. 167. Consider if "either" should be replaced by "combinations of".

18) We have amended the text accordingly.

OK

Fig. 2a and b could be combined into a single panel as the order of the experiments is the same.

19) Both panels are qualitative (a) and quantitative (b) analyses of the same experiment and are split into separate panels for clarity in line with other figures.

OK, your choice.

Fig. 3a. Consider colouring the monomer in a more informative way, e.g. to show domains, secondary structure etc. The all-purple cartoon doesn't say much. Alternatively, 3a can be left out entirely as all of this can be seen from 3b.

20) The colouring of the toxin has been altered to depict secondary structure elements.

OK, better.

Fig. 3d. Please include the electrostatic potential scale (in kBT_e-1c or RT_e-1c) from red to blue from APBS.

21) The requested potential scale has been included for reference.

The included scale only shows that red is - and blue is +. You need to include the actual scale in a suitable unit so that one can see what max. blue (and max. red) correspond to biophysically.

Fig. 4f. There is no SEM on the last (orange) bar. If the errors are that small, is the difference between red and orange really not significant?

22) Adjusted p-values were determined to be 0.9997, 0.9954, and 0.4964 when comparing changes in T_m when incubated with UTP vs ATP, CTP, and GTP, respectively. As such, no statistical significance in the changes to T_m were detected.

Really, a bit surprising, I think. But OK now that you have the data points plotted, I guess.

L. 321. What does Mr mean?

23) Mr referred to molecular weight; we have amended this for clarity.

OK

Ref: Nature Communications manuscript NCOMMS-24-12797-T

Title: Inducible auto-phosphorylation regulates a widespread family of nucleotidyltransferase toxins

Response to Editor and Reviewers

We would like to thank the Editor and Reviewers for their positive comments and constructive criticism on our revised manuscript. Point-by-point responses to Editorial and Reviewer comments from the first round of revisions are listed below in blue, whilst responses to comments from the second round of revisions are shown in orange. Where necessary, changes made to the original manuscript and are shown in red. In some instances, changes have been made to figures but these are not altered in color, in order to maintain consistency within the figure.

Reviewer #1 (Remarks to the Author):

The authors have adequately responded to the remarks and suggestion provided during the first review.

>Thank you for these positive comments and for your suggestions, we feel they have strengthened the manuscript.

Reviewer #2 (Remarks to the Author):

The authors have addressed all the concerns and suggestions that I had made in the original review. I am satisfied with the response. I would like to congratulate the authors on a nice discovery and interesting findings.

>Thank you for the praise, we are grateful for the suggestions made and feel as though the manuscript is stronger as a result.

[Please note below includes comments and responses from first round, onto which the Reviewer added second round comments]

Reviewer #4 (Remarks to the Author):

Overall, the manuscript by Arrowsmith et al. presents a convincing and highly intriguing account of how antitoxin binding induces phosphorylation in a widespread family of NTase toxins and repurposes the toxin active site from nucleotide transfer towards auto-phosphorylation in order to control activity. The study elegantly combines in vitro functional studies using a wide range of biochemical assays with structure prediction and experimental structure determination as well as molecular modelling to arrive at a cohesive and convincing molecular model. Overall, the manuscript is highly interesting and is suitable for publication following careful revision. However, in the current version, several of the figure panels are somewhat preliminary and could be improved. My main concerns therefore relate to presentation of the data as outlined below.

Thank you for these positive comments.

Several of the figure legends are too brief and do not provide the required information to interpret

the data, e.g. 1a, 1d and S5c, see specific comments below. Please carefully check that all figure elements are described in all legends including all lanes/bars/labels.

1) We thank the Reviewer for this feedback. Figure legends have been checked to ensure the information provided allows the reader to fully interpret the data.

OK

L. 180. I am not convinced that you can conclude that Mg²⁺ is not required. Have you considered that it could be carried along by one of the protein or nucleotide components, consistent with the observation that you see increased activity with added Mg²⁺? In fact, I would be surprised if Mg²⁺ was not required.

2) We thank the Reviewer for their insight here. We repeated phosphorylation experiments by first pre-incubating protein components with EDTA for 1 hour, then buffer exchanged to remove EDTA. CTP was then added to samples, either in the absence or presence of MgCl₂. In the absence of MgCl₂, no phosphorylation could be detected, confirming Mg is indeed essential for phosphorylation:

'Having identified increased phosphorylation activity in the presence of MgCl₂, we sought to establish whether magnesium was essential for MenT₁ phosphorylation. MenT₁ and MenA₁ were first pre-incubated with 5 mM EDTA for 1 hour to facilitate chelation of protein-bound metals prior to overnight dialysis. Supplementation of protein mixtures with 1 mM CTP alone failed to induce a change in the mass of MenT₁ (**Supplementary Fig. 2I**). In contrast, supplementation with 1 mM CTP and 10 mM MgCl₂ produced an increase in mass of 81 Da corresponding to the addition of a phosphate (**Supplementary Fig. 2J**), confirming magnesium is essential for phosphorylation activity.'

OK, good to have this clarified.

For bar charts, I believe it is recommended to all measured data points as individual markers if n<10, not just mean +/-SEM. Since you do not have that many repetitions, I suggest you follow that standard for all bar charts.

3) All individual points have been shown in line with standard practice for data where n<10.

OK, this is perhaps more of an esthetic comment, but it's hard to see the red crosses, especially when the bar outline is red as well, like in Fig. 4e and f. I think the bar charts are over-complicated by too many colours that are not needed. For example, you don't need each experiment to be represented by a different colour if there is a label below. Also, I would streamline fonts and their sizes as well as colours and styles more across the figures as currently each new bar chart has a different style from the previous one.

>We appreciate the advice regarding formatting. Care and consideration has been taken to ensure consistency across graphs and charts. For example, all axes and labels are the same size and use the same font, whilst colors for graphs such as those in Fig. 4 match respective colors for nucleotides with which they correspond, to allow readers to easily compare between panels. We have amended SEC profiles within Fig. 5 so that they all share the same color scheme.

Overall, I am disappointed with the quality of the SDS-PAGE (PhosTag) gels, most of which have run very unevenly. Sometimes, like in Fig. 2a and S6a this even makes it hard to interpret the data. What is the reason for this? We've done Phos-tag gels and I believe they are regular SDS-PAGE gels with an extra reagent added, so it should be straightforward to make them run more smoothly with commercial gels. Consider repeating some of the most important ones (like those in the main figures) for a better and more convincing overall presentation of the paper.

4) We argue that the gels can still be readily interpreted and that the conclusions are supported by our MS data. We did try to re-run these experiments, but have found it very hard to procure additional gels. Whilst we accept this is not the best response, having ordered additional gels three months ago to repeat these experiments, they still have not arrived from Japan. We have decided to keep the gel images we have.

I agree that this is not the best response and I think the gel quality detracts from the strength of the conclusions drawn from Fig. 2a and 2e. Given that you have two data points, could you include the other gels in Supplementary Information for transparency? I think it's a bit problematic to integrate gels of this quality, especially for the L14R/V19R mutant, in which it's hard to see the phosphorylated protein band.

>The phosphorylated band is hard to see because there is very little phosphorylation by the mutant. We show these data (Fig. 2F) and state objectively that, "MenA₁ L14R/V19R caused reduced levels of MenT₁ phosphorylation compared to MenA₁ WT", also referring the reader to our quantitative mass spec experiments in Supplementary Fig. 4C, which supports the results of Phos-Tag SDS-PAGE in that there was no phosphorylation detected by mass spec.

Figure 2f. How can the mean of a integrated band intensities be zero with a large SEM (WT + bar)? An integrated band intensity cannot be negative, so if the SEM is large (about +/- 20% here) then the mean must be positive and >0. Please check the data once more.

5) We thank the Reviewer for spotting this mistake; this dataset had been incorrectly normalised. Data now reflects raw density, which matches all other lanes.

OK, more clear, thanks.

L. 223. For the RMSD, it appears that all atoms were used (1127)? I don't think this the best way to reveal folding changes (side chains can be floppy and are affected by crystal contacts). Instead, I would calculate RMSD using C-alpha atoms only and compare the values for the entire protomer to that calculated from the loop region only to show that the loop moves upon phosphorylation.

6) Alignments were repeated to show a) sequence-independent overlay of entire protomers, b) alignment of C-alpha atoms across entire protomers, and c) alignment of C-alpha atoms across the loop region from either protomer:

'Sequence-independent structural overlay of MenT₁ and MenT₁-p returned a root mean square deviation (RMSD) of 0.446 Å (1127 atoms), indicating a near identical topology of core domains and folds (Fig. 3C). Alignment of each structure via C-alpha atoms only, returned an RMSD of 0.370 Å (1127 atoms), whereas aligning the C-alpha atoms of just the respective T39 loops returned an RMSD of 0.778 Å (7 atoms), indicating a larger shift between these specific loops.'

Not sure why you need the entire promoter overlay if the point is just that the fold is the same? Also, in the text above, how can you have the same number of atoms (1127) for both all-atom and C-alpha-only overlays?

>We agree that the all-atom overlay is made redundant by aligning the C-alpha atoms of each protomer and have removed this from the text. Similarly, we thank the reviewer for spotting this error in the alignment, the typo has been corrected. The text now reads as follows:

Alignment of each structure via C-alpha atoms only returned an RMSD of **0.410 Å (160 atoms)**,

indicating a near identical topology of core domains and folds (**Fig. 3C**). In contrast, aligning the C-alpha atoms of the respective T39 loops returned an RMSD of 0.778 Å (7 atoms), indicating a larger shift between these specific loops.

L. 248. Isn't the difference in melting point just 1.4° between CTP and the other nucleotides? From the figures, it appears that the thermal shift is 3.8° for CTP and 2.4° for the others, so a 1.4° difference between them, please check this.

7) We thank the Reviewer for spotting the error, we have adjusted the text accordingly to rectify this mistake.

OK

L. 272. Is there no structure of a homologous protein with nucleotide bound that you can use for comparison of the binding mode? It would be nice to confirm the docking poses. Also, I find it very odd that you observe no base-specific interactions given that the NTase prefers CTP. Can you comment on this?

8) Whilst the manuscript was in review, another group published the crystal structure of MenT3 bound to CTP (<https://doi.org/10.1093/nar/gkae177>). We have compared the MenT3 ATP MD pose to that of the solved structure for reference, and observed a close match (**Supplementary Fig. 9E-F**, lines 452-465). With regards to the MenT1 NTP docking poses, we have added an additional reference to a related NTase, Cid1, which forms the majority of its protein-ligand contacts through interactions with the triphosphate tail of each NTP. Finally, we feel that scrutinization of differences in NTP binding modes for a structural explanation for the observed preference of MenT1 for CTP as an NTase substrate does not fit with the scope of our research, as the toxin exhibits no preference for phosphorylation substrates, which our docking models support. It is likely we will need to also consider the tRNA target to help analyze CTP preference for the NTase activity, in a future study.

OK

Fig. 5efg. I don't believe it's appropriate to have MW on the x-axis of a SEC trace. You should always have elution volume (the true variable) and then indicate with arrows the elution volumes and MW your standard proteins. Also, it looks like MenT-p alone shifts compared to the unphosphorylated versions, can you comment on this?

9) SEC traces have been amended accordingly. With regards to MenT1-p elution, we note that the calculated R_{st} value of MenT1-p from its crystal structure is smaller than for MenT1 (see **Supplementary Fig. 8**), which may explain the difference in the elution trace for MenT1-p relative to MenT1 WT/T39A.

OK, but if you have done MW standards for the SEC column you could put those as points on the x axis (with an arrow pointing to their elution volume) to let the reader assess the apparent mass of your complexes.

>SEC overlays have dashed lines to illustrate the expected elution volumes of each species based on their hydrodynamic radius, which provides a better estimate for elution from the column than molecular weight alone. As an additional aid, we have modified Supplementary Fig. 8 by adding two additional panels (A and B) to allow for comparisons between respective co-incubation samples and MW standards, and have also updated Supplementary Figs. 8E-G to now also show molecular weight ratios.

Fig. S5d. Please include the unbiased Fo-Fc difference map from before modelling the phosphate group into the structure. Refined 2Fo-Fc is biased by the model and could in principle show artificial phosphate group density.

10) We have included the unbiased difference map to reflect the presence of missing density prior to modelling of the phosphate (see **Supplementary Fig. 5E**).

OK, looks convincing.

Figure 1a. It is not clear from the figure if it is the N-terminal or C-terminal parts of the toxins that are conserved. I would suggest that you use dashed lines going across the isoforms or similar colours of domains to indicate conserved parts now that the figure is drawn to scale anyway. In the legend, what does "original and revised nomenclature" refer to?

11) As a visual aid, we have aligned each of the four MenT toxins to illustrate conserved motifs, see Fig. S1a. We kindly direct the Reviewer to Dy et al 2014 *NAR* Fig. 8a, in which the authors illustrate the four conserved DUF1814 motifs and where they are located: <https://academic.oup.com/nar/article/42/7/4590/2436634?login=true>. We have also clarified in the figure legend that original and revised nomenclature refers to original gene identifiers and revised nomenclature of gene products.

I would still prefer to have Figure 1a show this alignment by having the domain boxes somehow related by vertical lines, but this is my personal taste and I accept that this is at the authors' discretion.

>The purpose of Fig. 1A is to solely to convey the relative sizes of each TA pair and to show which proteins belong to which superfamilies. As we have not mentioned N- or C- terminus conservation within the manuscript, and because conserved DUF1814 motifs are scattered throughout the structures of respective toxins, we suggest this would not improve the figure.

Figure 1b. I would be more cautious about listing the estimated MW for the SEC experiments. SEC elution volumes depend on hydrodynamic radii, which are only app. corresponding to MW, so it makes no sense to list MW so precisely (5 digits). Also, please include a gel showing whether (or not) the two proteins bind and co-elute with a tag on one protein in the three cases.

12) Values shown were calculated (known) MW values for respective proteins, however, for the sake of clarity as this was not explicitly stated, SEC traces have been adjusted accordingly to simply show elution volumes for each sample. We refer the Reviewer to Cai et al 2020 *Sci Adv* Fig. S4 for our previously published co-elution gels <https://www.science.org/doi/10.1126/sciadv.abb6651>.

OK

Figure 1d. What does ATc +/- refer to? Also, the blue colour is very faint, so I would like to ask you to include a positive control where there is a clear repression of the lac promoter.

13) The figure legend has been updated for additional clarity with regards to the use of the pGMC inducer anhydrotetracycline (ATc). As requested, we included a positive control in β -galactosidase assays to show clear repression of the *parDE1* promoter by the ParDE1 complex. Please also note this panel has been moved to **Supplementary Fig. 1F**, and replaced with the liquid growth quantifications of LacZ activity that were originally in the Supplementary Materials, as we felt the quantitative data are more impactful having now been enhanced with the addition of a positive control and therefore should be in the main figures.

'We examined whether there was **also** any transcriptional regulation by cloning the 1000 bp region immediately upstream of the *menAT1* transcriptional start site into the promoterless *lacZ* fusion construct pJEM15²⁸, and quantified β -galactosidase activity in *M. smegmatis* during co-induction of MenA₁, MenT₁, or both together (**Fig. 1D; Supplementary Fig. 1F**), using ParDE1 as a positive control for transcriptional repression.'

OK

L. 132. It should be mentioned how you can express the isolated toxins given that they inhibit translation? This is not possible for the majority of TA toxins.

14) We have added additional detail to lines 134-137:

'Despite demonstrable toxicity of MenT NTases high yields of each lone toxin homologue (2- 5 mg/L) can be obtained following recombinant protein expression in *E. coli* using rich media, which we previously suggested may be a result of elevated tRNA target levels in *E. coli* relative to *M. tuberculosis*.'

OK

L. 134. Please include a gel showing the stoichiometry as this cannot be seen from the SEC alone (see above).

15) We have amended **Supplementary Fig. 1** to include calculated/observed Molecular Weight and Stokes radius ratios for MenT1 lone and co-expression samples to reflect that the co-expression peak matches the expected Rst of the heterotrimer.

I don't think this adds much. I maintain that an SDS-PAGE gel would be very useful to support the proposed stoichiometry.

>We recognize that an SDS-PAGE would confirm the presence of two components but this is redundant as it has been confirmed through crystallography. The SDS-PAGE would not give a quantifiable stoichiometry. We have opted not to perform this experiment.

L. 145. MS measures molecular mass, not weight.

16) We have amended the text accordingly and thank the Reviewer for spotting this mistake.

OK

L. 162. The last statement is very blunt - I don't think you can exclude all transcriptional regulation based on this single experiment.

17) We have altered our concluding statement to be more conservative.

OK

L. 167. Consider if "either" should be replaced by "combinations of".

18) We have amended the text accordingly.

OK

Fig. 2a and b could be combined into a single panel as the order of the experiments is the same.

19) Both panels are qualitative (a) and quantitative (b) analyses of the same experiment and are split into separate panels for clarity in line with other figures.

OK, your choice.

Fig. 3a. Consider colouring the monomer in a more informative way, e.g. to show domains, secondary structure etc. The all-purple cartoon doesn't say much. Alternatively, 3a can be left out entirely as all of this can be seen from 3b.

20) The colouring of the toxin has been altered to depict secondary structure elements.

OK, better.

Fig. 3d. Please include the electrostatic potential scale (in $k_B T e^{-1} c$ or $R T e^{-1} c$) from red to blue from APBS.

21) The requested potential scale has been included for reference.

The included scale only shows that red is - and blue is +. You need to include the actual scale in a suitable unit so that one can see what max. blue (and max. red) correspond to biophysically.

>We have amended the figure and its legend to now include the appropriate scale with the relevant numerical values for charges and defined the units within the legend.

Fig. 4f. There is no SEM on the last (orange) bar. If the errors are that small, is the difference between red and orange really not significant?

22) Adjusted p-values were determined to be 0.9997, 0.9954, and 0.4964 when comparing changes in T_m when incubated with UTP vs ATP, CTP, and GTP, respectively. As such, no statistical significance in the changes to T_m were detected.

Really, a bit surprising, I think. But OK now that you have the data points plotted, I guess.

L. 321. What does Mr mean?

23) Mr referred to molecular weight; we have amended this for clarity.

OK